# *SynAdapt*: Learning Adaptive Reasoning in Large Language Models via Synthetic Continuous Chain-of-Thought

## Abstract

While Chain-of-Thought (CoT) reasoning improves model performance, it incurs significant time costs due to the generation of discrete CoT tokens (DCoT). Continuous CoT (CCoT) offers a more efficient alternative, but existing CCoT methods are hindered by indirect fine-tuning, limited alignment, or inconsistent targets. To overcome these limitations, we propose *SynAdapt*, an innovative efficient reasoning framework. Specifically, *SynAdapt* generates the synthetic CCoT to serve as a precise and effective alignment target for LLMs. This synthetic CCoT explicitly guides the LLM to learn CCoT and derive accurate answers directly. Furthermore, relying solely on CCoT is insufficient for solving hard questions. To address this, *SynAdapt* integrates a difficulty classifier that leverages both question context and CCoT to identify hard questions. CCoT can effectively help identify hard questions after some brief reasoning. We then adaptively prompt the LLM to re-think these hard questions for improved performance. Extensive experimental results across various benchmarks from different difficulty levels strongly demonstrate the effectiveness of our method, achieving the best accuracy-efficiency trade-off. [1]

## 1 Introduction

Chain-of-Thought (CoT) reasoning (Kojima et al., 2022; Wei et al., 2022; Zhou et al., 2022) has shown remarkable potential in enhancing the problem-solving capabilities of Large Language Models (LLMs) for complex tasks (Guo et al., 2025; Yang et al., 2025; OpenAI, 2025). By decomposing problems into sequential steps, CoT allows LLMs to derive correct answers step-by-step. However, a major drawback of CoT is its high computational cost due to the generation of numerous tokens, which leads to substantial time consumption (Yu et al., 2024; Yeo et al., 2025). While this cost is often acceptable in **accuracy-sensitive scenarios**, such as AI for Science (AI4S) (Lu et al., 2024) where accuracy is paramount, it becomes problematic in **efficiency-sensitive scenarios**. For instance, in embodied intelligence, real-time human-computer interaction necessitates highly efficient reasoning to ensure a satisfactory user experience (Li et al., 2024a). Consequently, a critical challenge emerges: how to reduce the length of generated CoT while preserving its effective reasoning capabilities.

Existing efficient reasoning approaches mainly involve fine-tuning or direct prompting LLMs to reduce the number of COT steps (Arora & Zanette, 2025; Munkhbat et al., 2025; Xu et al., 2025a). However, the remaining CoT steps still involve numerous discrete natural language tokens, which we refer to as **DCoT**. As noted by Li et al. (2024b) and Lin et al. (2024), most of these verbalized tokens are mainly for communication and carry unnecessary linguistic details that do not contribute to the core reasoning process. One promising approach is fine-tuning LLM to replace DCoT with a more compact and continuous CoT representation, known as **CCoT** (Pfau et al., 2024; Goyal et al., 2023). During reasoning, CCoT retains the hidden state of the LLM and skips generating the one-hot token ID, allowing it to store more information than just a single token (Zhu et al., 2025).

Nonetheless, fine-tuning LLM to learn CCoT reasoning effectively remains challenging. Coconut (Hao et al., 2024) gradually fine-tunes the LLM to replace DCoT with CCoT using a curriculum learning strategy (Deng et al., 2024). However, as shown in Figure 1, it lacks explicit alignment

---

[1] We have released all our code and dataset in the supplementary materials for better review.

Figure 1: Comparisons between our *SynAdapt* and the other CCoT-based baselines. These baselines either train CCoT indirectly, provide only single-position alignment, or apply full alignment with incoherent targets.

between DCoT and CCoT, which limits its ability to effectively learn from the original DCoT. CODI (Shen et al., 2025b) introduces explicit alignment between the last token hidden state of DCoT and the final hidden state of CCoT, but ignores alignment for other intermediate tokens. CompressCoT (Cheng & Van Durme, 2024) attempts to identify a subset of important tokens from DCoT, whose length matches CCoT, and aligns the full CCoT with the hidden states of these tokens. However, selecting only several isolated DCoT tokens leads to incoherence in the reasoning process. This leads to significant performance degradation in CCoT learning.

To overcome these limitations, we propose a novel efficient reasoning framework called *SynAdapt*, which helps LLM learn **Adapt**ive reasoning through **Syn**thetic CCoT. Our approach begins by generating a synthetic CCoT to serve as a comprehensive alignment target. Specifically, we initialize a random CCoT, fix the LLM, and iteratively optimize the random CCoT into a synthetic CCoT to guide the LLM towards correct answers. The synthetic CCoT thereby serves as a better alignment target than only using several isolated and incoherent tokens from the original DCoT. During fine-tuning, we apply the full alignment using the synthetic CCoT, as shown by Figure 1. This strategy helps LLM learn the full CCoT rather than only the last one. Notably, we fine-tune the LLM to iteratively refine a meaningless draft to obtain the CCoT, rather than generating CCoT autoregressively. This approach is more efficient (Jiang et al., 2025) and can boost the reasoning ability of LLM by iterative refinement (Saunshi et al., 2025; Yu et al., 2025).

Moreover, according to the information theory (Nalewajski, 2011), compressing DCoT into the dense CCoT inevitably leads to information loss and increases the complexity of solving hard questions (Koehn & Knowles, 2017). We provide an example in Figure 5 of the Appendix. To address this, we train a difficulty classifier that assesses question difficulty based on both the question itself and the CCoT. And then prompt the LLM to re-think hard questions using discrete CoT tokens for improved accuracy. While CCoT may not be sufficient to solve these hard questions, it can help the classifier effectively identify them. Some hard questions resemble simpler ones and can only be distinguished through the brief reasoning captured by CCoT. We also present an illustrative example in Figure 6 of the Appendix.

We evaluate our method across various benchmarks with different difficulty levels, including GSM8K, MATH500, AMC23, AIME24, and AIME25. By dynamically adjusting the ratio of re-think hard questions, our method demonstrates adaptability in both accuracy-sensitive and efficiency-sensitive scenarios. Comprehensive experimental results demonstrate that our method outperforms other baselines in both scenarios, achieving an optimal accuracy-efficiency trade-off. We further assess identification performance of our difficulty classifier, showing its superior performance compared to other baselines. In addition, we evaluate the generalization capacity of our method across broader domains, such as scientific QA and coding, as well as under different LLM backbones. The main contributions of this paper are as follows:

- We propose a novel efficient reasoning framework that generates synthetic CCoT, providing a better full alignment target to help LLMs learn CCoT more effectively.

- We introduce a difficulty classifier that more effectively distinguishes hard questions by considering both the question and the CCoT, enabling adaptive re-thinking for improved accuracy.

- Extensive experimental results strongly demonstrate the effectiveness of our framework, achieving the best accuracy-efficiency trade-off.

## 2 RELATED WORK

In this section, we introduce the mainstream related work on efficient reasoning in the LLMs, which can be mainly categorized into three types: SFT-based methods, RL-based methods, Prompt-based methods, and CCoT-based methods.

**SFT-based methods** either discard the CoT entirely or dynamically compress the CoT in the training data. And then they apply supervised fine-tuning (SFT) on these compressed data to help LLM learn to reduce generation length. While these methods are effective in shortening the generated output, they may ignore some crucial details of the original CoT during fine-tuning, leading to significant performance degradation (Yu et al., 2024; Ma et al., 2025b; Munkhbat et al., 2025; Xia et al., 2025; Kang et al., 2025). **RL-based methods** primarily design length penalties to prevent the model from generating excessively long CoT. While these methods can reduce reasoning length without sacrificing LLM performance, they require substantial resources for repeated data sampling to LLM training. Additionally, the reduction in length is limited and may not be suitable for efficiency-sensitive scenarios, where minimizing generation length is crucial (Arora & Zanette, 2025; Luo et al., 2025; Yeo et al., 2025; Aggarwal & Welleck, 2025; Shen et al., 2025a). **Prompt-based methods** explicitly add length constraint instructions in the prompt for guiding LLM to reduce generation length. Although these approaches are low-cost, their impact on length reduction is limited. LLMs still tend to generate long, redundant reasoning CoTs, especially for those hard questions (Renze & Guven, 2024; Xu et al., 2025a; Lee et al., 2025; Han et al., 2024).

Instead of reasoning by numerous redundant tokens, **CCoT-based methods** aim to compress the reasoning steps by replacing the original discrete CoT (DCoT) with Continuous CoT (CCoT) in the latent space. However, these methods often suffer from significant performance drops. This is mainly because they either don't explicitly align CCoT with DCoT or only use parts of the DCoT to conduct alignment. These weak alignment signals can not effectively help LLM learn CCoT reasoning, leading to the performance degradation (Hao et al., 2024; Xu et al., 2025b; Shen et al., 2025b; Cheng & Van Durme, 2024). Due to the limited space, a detailed introduction of the above related works are shown in Appendix B.

## 3 METHODOLOGY

In this section, we present the details of our *SynAdpat* framework, which consists of two stages: the fine-tuning stage and the inference stage, as shown in Figure 2. During the fine-tuning stage, we first **generate the synthetic CCoT** by optimizing a randomly initialized one. The optimization goal is to ensure that the LLM generates the correct answer when using the synthetic CCoT. After generation, we fine-tune the LLM to learn CCoT by **utilizing the synthetic CCoT as the alignment target**. Specifically, the LLM is trained to iteratively refine a draft CCoT until it aligns with the pre-generated synthetic CCoT Additionally, we **train a difficulty classifier** that assesses a question's difficulty based on both the question itself and its corresponding CCoT.

During the inference stage, the fine-tuned LLM generates the CCoT for the given question. This generated CCoT, along with the original question, is then fed into the difficulty classifier to **distinguish between easy and hard questions**. For easy questions, the LLM directly generates the answer based on the CCoT, ensuring high efficiency. For hard questions, we discard the CCoT and prompt the LLM to re-think the question step by step, ensuring higher accuracy. More details of the training stage and the inference stage are presented in Section 3.1 and Section 3.2, respectively.

### 3.1 TRAINING STAGE

**Synthetic CCoT Generation.** To provide a more effective alignment target to learn CCoT representation during fine-tuning LLM, we firstly generate the synthetic CCoT before fine-tuning.

As shown in the upper-left part of Figure 2, for each question $Q$, we randomly initialize a synthetic CCoT $Z_{\text{syn}}$ with a fixed length $m$. We then concatenate $Q$ with $Z_{\text{syn}}$ and an end-of-think token to form $[Q, Z_{\text{syn}}, \text{eot}]$. Given that a well-constructed CCoT should guide the LLM to predict the correct answer based on the question and CCoT, we make $Z_{\text{syn}}$ trainable and optimize it by minimizing the

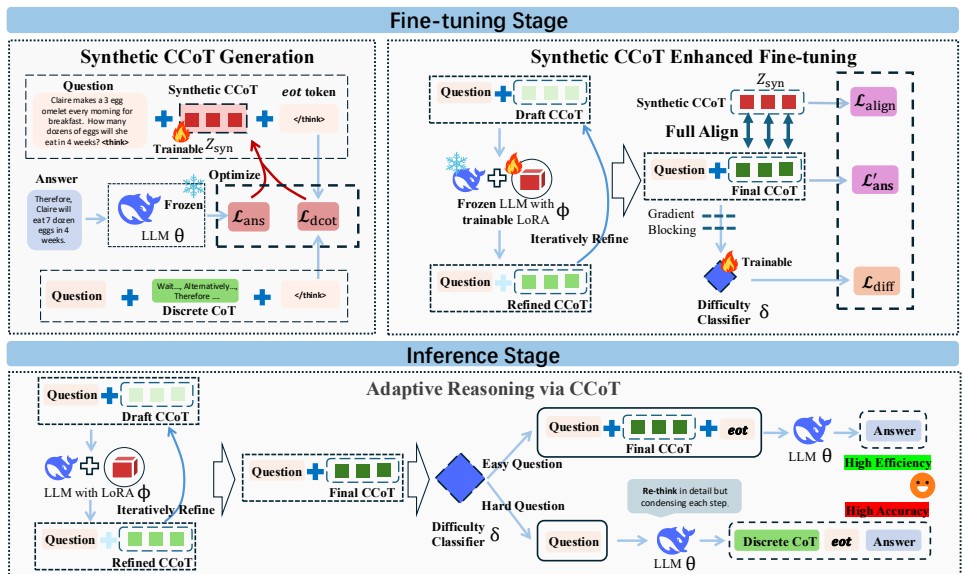

Figure 2: Our *SynAdapt* framework consist of two stage. (1). In **Synethetic CCoT Generation**, we first generate the synthetic CCoT $Z_{\text{syn}}$ for each question. And then in **Synethetic CCoT Enhance Fine-tuning**, $Z_{\text{syn}}$ serves as the full alignment target. By using $Z_{\text{syn}}$, we fine-tune the LLM $\phi$ to effectively learn CCoT, enabling iterative refinement of a randomly initialized draft CCoT. Additionally, we train a **difficulty classifier** $\delta$ to assess question difficulty based on both the question and the generated CCoT. (2). During the inference stage, we use the fine-tuned LLM $\phi$ to iteratively refine and generate the final CCoT, while the difficulty classifier $\delta$ determines the question difficulty. **For easy questions**, the LLM directly generates the output, and **for hard questions**, it is prompted to re-think in order to generate the correct answer.

following loss:

$$\mathcal{L}_{\text{ans}} = -\frac{1}{L_a} \sum_{i=1}^{L_a} \log \mathcal{P}_\theta(A_i | Q, Z_{\text{syn}}, \text{eot}, A_{<i}), \tag{1}$$

where $L_a$ is the length of the answer $A$, $A_i$ denotes the $i$-th token of $A$, and $\theta$ represents the parameters of the LLM.

Moreover, to prevent overfitting during CCoT optimization, we additionally align the hidden state of the eot token when using the synthetic CCoT with that obtained when using DCoT. Assuming $\mathbf{h}_{\text{eot\_syn}}^l$ is the hidden state of the eot token at the $l$-th layer of the LLM when provided with synthetic CCoT $Z_{\text{syn}}$ and $\mathbf{h}_{\text{eot\_dcot}}^l$ is that when provided with DCoT, the alignment loss is defined as:

$$\mathcal{L}_{\text{dcot}} = \frac{1}{L} \sum_{l=1}^{L} \left\| \mathbf{h}_{\text{eot\_syn}}^l - \mathbf{h}_{\text{eot\_dcot}}^l \right\|_1, \tag{2}$$

where $L$ is the total number of layers in the LLM. After optimizing using both $\mathcal{L}_{\text{ans}}$ and $\mathcal{L}_{\text{dcot}}$, we obtain the high-quality synthetic CCoT $Z_{\text{syn}}$, which serves a similar function to DCoT but is represented in a denser, continuous format. These $Z_{\text{syn}}$ can serve as valuable alignment targets during fine-tuning LLM to learn CCoT.

**Synthetic CCoT Enhanced Fine-tuning.** As demonstrated by Saunshi et al. (2025); Yu et al. (2025), iteratively looping an LLM can significantly enhance its reasoning capabilities and refine outputs. Inspired by this, we fine-tune the LLM to iteratively refine the CCoT from a draft in a looping manner instead of generating it autoregressively.

As shown in Figure 2, we concatenate the question $Q$ with a draft CCoT $Z_{\text{draft}}^0$. The $Z_{\text{draft}}^0$ is initialized as the embedding of a repeated meaningless token sequence (i.e., <T>..<T>), with a fixed length of $m$. We input the $Z_{\text{draft}}^0$ into LLM and use the corresponding output hidden state as the refined one.

The iterative refinement process can be formulated as:

$$Z_{\text{draft}}^i = f_\phi(Q, Z_{\text{draft}}^{i-1})[L_q :], \tag{3}$$

where $Z_{\text{draft}}^i$ is the CCoT after refining $i$ iterations, $L_q$ is the length of the question $Q$, $\phi$ represents the fine-tuned LLM with a trainable LoRA module and $f_\phi(\cdot)$ returns the output hidden state from $\phi$. After $k$ refining iterations, we obtain the final CCoT $Z_{\text{final}} = Z_{\text{draft}}^k$. We explicitly align the full $Z_{\text{final}}$ with the synthetic CCoT $Z_{\text{syn}}$ and compute the $\mathcal{L}_{\text{align}}$ loss as:

$$\mathcal{L}_{\text{align}} = \|Z_{\text{final}} - Z_{\text{syn}}\|_1. \tag{4}$$

Moreover, $Z_{\text{final}}$ should also guide the initial LLM to generate the correct answer. Therefore, we compute an additional losses, similar to Equation 1, as shown below:

$$\mathcal{L}'_{\text{ans}} = -\frac{1}{L_a} \sum_{i=1}^{L_a} \log \mathcal{P}_\theta(A_i | Q, Z_{\text{final}}, \text{eot}, A_{<i}), \tag{5}$$

$$\mathcal{L}_{\text{refine}} = \mathcal{L}_{\text{align}} + \mathcal{L}'_{\text{ans}}, \tag{6}$$

where $\theta$ represents the initial LLM without the LoRA module. The $\mathcal{L}_{\text{refine}}$ loss fully utilizes the alignment information from $Z_{\text{align}}$. After training using $\mathcal{L}_{\text{refine}}$, the fine-tuned LLM $\Phi$ effectively learns to iteratively refine the draft CCoT, ultimately generating the final CCoT to replace the original redundant DCoT.

**Difficulty Classifier Training.** Additionally, we train a difficulty classifier $\delta$, composed of two MLP layers, to distinguish between hard and easy questions. It takes both the question itself and the CCoT as input. Specifically, we construct question pairs $\langle Q_c, Q_r \rangle$ based on existing difficulty labels from the DeepMath dataset (He et al., 2025). $Q_c$ is a hard question and $Q_r$ is an easy question. Next, we input $Q_c$ and $Q_r$ to the fine-tuned LLM $\phi$ to obtain the corresponding CCoT $Z_{\text{final}}^c$ and $Z_{\text{final}}^r$. Then we concatenate $Q_c$, $Z_{\text{final}}^c$ and one eot token and input to the initial LLM to obtain the output hidden state of eot as:

$$\mathbf{h}_{\text{eot\_final}}^c = f_\theta(Q_c, Z_{\text{final}}^c, \text{eot})[-1], \tag{7}$$

where $f_\theta$ represents the output hidden state from the initial LLM $\theta$ and $\mathbf{h}_{\text{eot\_final}}^c$ denotes the output hidden state of the eot token. Considering the attention mechanism of LLM, $\mathbf{h}_{\text{eot\_final}}^c$ can fully capture the information in $Q_c$ and $Z_{\text{final}}^c$. Similarly, we compute the $\mathbf{h}_{\text{eot\_final}}^r$ for the easy question $Q_r$. We train the difficulty classifier $\delta$ according to the following loss:

$$\mathcal{L}_{\text{diff}} = -\log \sigma(f_\delta(\mathbf{h}_{\text{eot\_final}}^c) - f_\delta(\mathbf{h}_{\text{eot\_final}}^r)), \tag{8}$$

where $f_\delta(\cdot)$ denotes the difficulty level predicted by $\delta$. $\mathcal{L}_{\text{diff}}$ encourages the classifier to give higher score for hard question $Q_c$ and lower scores to easy ones $Q_r$. By utilizing additional information from the CCoT, the classifier $\delta$ can more effectively distinguish between hard and easy questions.

## 3.2 INFERENCE STAGE

**Adaptive Reasoning via CCoT.** During the inference stage, we concatenate the question with a draft CCoT and utilize the fine-tuned LLM $\phi$ to iteratively refine the draft CCoT to obtain the final CCoT. And then we utilized the difficulty classifier to assign the difficulty score based on both the question and the CCoT. Questions with a difficulty score below the threshold $\tau$ are considered easy, while those above are regarded as hard.

**For easy questions**, we just append a eot token after the CCoT and prompt the base LLM $\theta$ to directly output answer. The generated CCoT effectively replaces the original discrete CoT reasoning process, which often contains numerous tokens and is time-consuming to generate, thereby achieving higher efficiency. However, compressing DCoT into CCoT inevitably leads to information loss (Nalewajski, 2011). And as shown by Hao et al. (2024), relying solely on CCoT is insufficient **for hard questions** and may even lead to incorrect answer. Therefore, we discard the generated CCoT and prompt the LLM to re-think the question via discrete CoT, using a more detailed reasoning process to generate the correct answer. Additionally, inspired by Xu et al. (2025a), we explicitly prompt the LLM to condense each reasoning step, achieving a better trade-off between accuracy and efficiency.

Moreover, we can dynamically adjust the threshold $\tau$ to control the ratio of re-thinking. This allows our method to simultaneously adapt to both accuracy-sensitive and efficiency-sensitive scenarios according the specific requirements of the real application. All our used prompts are provided in Appendix H.

## 4 EXPERIMENTS

In this section, we conduct comprehensive experiments to demonstrate the effectiveness of our *SynAdapt* and address the following four key research questions:

- **RQ1:** Can *SynAdapt* offer a better accuracy-efficiency trade-off compared to other efficient reasoning baselines in both accuracy-sensitive and efficiency-sensitive scenarios? (see section 4.1)

- **RQ2:** Does our difficulty classifier, which uses both the question and CCoT, can effectively distinguish between hard and easy questions? (see section 4.2)

- **RQ3:** How about the training efficiency of *SynAdapt*? (see section 4.3)

- **RQ4:** How well does *SynAdapt* generalize across more domains (Scientific QA/Coding), LLM backbones, and hyperparameters? (see Section section 4.4)

### 4.1 EVALUATION OF ACCURACY-EFFICIENCY TRADE-OFF

**Experimental Settings**   We use DeepMath (He et al., 2025) as the training set and evaluate our method and baselines on five widely adopted math-related benchmarks: AIME25, AIME24, AMC23, MATH500 (Lightman et al., 2023) and GSM8K (Cobbe et al., 2021). These datasets cover a diverse range of math questions across varying difficulty levels. As for the evaluation metrics, we report accuracy (**Acc**) and generation length (**Len**) to assess both performance and efficiency. Additionally, we introduce the **Relative Gain** metric (**Rel-G**) defined as:

$$\text{Rel-G} = \frac{\text{Acc}/\text{Acc}_{\text{raw}}}{\text{Len}/\text{Len}_{\text{raw}}}, \tag{9}$$

where $\text{Acc}_{\text{raw}}$ and $\text{Len}_{\text{raw}}$ denote the accuracy and generation length of the raw model, respectively. A higher Rel-G indicates a better trade-off between accuracy and efficiency. We also further evaluate our method on additional domains, including scientific QA and coding, in Section 4.4.

We adopt DeepSeek-R1-Distill-Qwen-7B as our raw model. We set the length of the synthetic CCoT to $m = 512$, and refining iterations for the draft CCoT to $k = 4$. The difficulty score ranges from 0 to 1. In accuracy-sensitive scenarios, we set threshold $\tau = 0.5$ to route difficult questions for re-thinking. In efficiency-sensitive scenarios, we set $\tau = 1.0$ to prompt the LLM to directly generate answers based on CCoT for higher efficiency. Further details on the datasets and implementation details are provided in Appendix C.1 and Appendix C.3 respectively.

**Compared Methods**   Here, we consider a broad range of existing efficient reasoning baselines, not limited to CCoT-based methods. We categorize these baselines into two scenarios, **accuracy-sensitive scenario** and **efficiency-sensitive scenario**, based on their different focuses.

In the accuracy-sensitive scenario, **CoT-FT** directly uses the full discrete CoT from the training data to perform supervised fine-tuning (SFT) for improving performance. **TokenSkip** (Xia et al., 2025) compresses the discrete CoT based on token importance and then applies SFT on the compressed CoT. **NoThinking** (Ma et al., 2025a) skips the SFT process and directly prompts the model to skip reasoning and directly generate the answer. **CoD** (Xu et al., 2025a) prompts the model to condense each reasoning step rather than skipping the reasoning process entirely. **TokenBudget** (Han et al., 2024) let the LLM to predict a token budget for each question in advance and prompts the model do not exceed the token budget during further generation.

In the efficiency-sensitive scenario, **NoCoT-FT** (Yu et al., 2024) discards the discrete CoT and performs SFT using only the answer to improve efficiency. **SelfTraining** (Munkhbat et al., 2025) applies best-of-$n$ sampling to extract the shortest correct CoT from the LLM and then fine-tunes the LLM on these CoT. **Coconut** (Hao et al., 2024), **CompressCoT** (Cheng & Van Durme, 2024), and **CODI** (Shen et al., 2025b) all belongs to CCoT-based methods, utilizing the CCoT to replace the DCoT for better efficiency. Coconut adopts a curriculum learning strategy to gradually internalize DCoT into CCoT. CompressCoT identifies key tokens in the DCoT and aligns the CCoT with their hidden states. CODI employs self-distillation, aligning the last token hidden state of CCoT with that of DCoT during training. More details of these compared method are provided in Appendix C.2.

**Main Results**   For the **accuracy-sensitive scenario**, as shown in the upper part of Table 1, our method with $\tau = 0.5$ outperforms all other baselines by achieving the second-highest average

| Methods | AIME25 | | AIME24 | | AMC23 | | MATH500 | | GSM8K | | Average | | |
|---|---|---|---|---|---|---|---|---|---|---|---|---|---|
| | Acc | Len | Acc | Len | Acc | Len | Acc | Len | Acc | Len | Acc ↑ | Len ↓ | Rel-G ↑ |
| Raw Model | 36.7 | 13348.6 | 53.3 | 14071.4 | 92.5 | 6315.7 | 93.2 | 4087.4 | 90.7 | 1110.8 | 73.3 | 7786.84 | 1.00 |
| *Accuracy-Sensitive Scenario* | | | | | | | | | | | | | |
| CoT-FT | 40.0 | 16427.3 | 40.0 | 15560.6 | 87.5 | 7049.1 | 88.6 | 3694.0 | 83.0 | 700.7 | 67.8 | 8686.4 | 0.83 |
| TokenSkip | 30.0 | 17811.3 | 36.7 | 14385.0 | 70.0 | 10030.8 | 78.4 | 16542.8 | 81.1 | 17165.5 | 59.2 | 15187.1 | 0.41 |
| NoThinking | 30.0 | 10623.6 | 40.0 | 11099.7 | 75.0 | 4143.6 | 82.4 | 1355.4 | 85.7 | 229.5 | 62.6 | 5490.4 | 1.21 |
| CoD | 40.0 | 10498.0 | 56.7 | 8488.5 | 80.0 | 2894.3 | 81.8 | 1591.1 | 84.2 | 286.2 | 68.5 | 4751.6 | 1.53 |
| TokenBudget | 36.7 | 15235.0 | 53.3 | 14897.7 | 82.5 | 5006.5 | 90.2 | 3186.8 | 86.9 | 573.0 | **69.9** | 7779.8 | 0.95 |
| *SynAdapt* ($\tau$=0.5) | 40.0 | 10198.3 | 56.7 | 8288.1 | 80.0 | 2881.6 | 82.4 | 1547.7 | 85.7 | 258.6 | 69.0 | **4694.8** | **1.58** |
| *Efficiency-Sensitive Scenario* | | | | | | | | | | | | | |
| NoCoT-FT | 13.3 | 637.0 | 10.0 | 1680.1 | 50.0 | 513.1 | 74.8 | 478.9 | 87.1 | 209.5 | 47.0 | 703.7 | 7.13 |
| SelfTraining | 10.0 | 671.6 | 10.0 | 772.7 | 55.0 | 627.0 | 71.6 | 397.0 | 85.1 | 207.6 | 46.3 | **535.2** | 9.10 |
| Coconut | 6.7 | 647.2 | 13.3 | 1692.5 | 52.5 | 548.0 | 76.2 | 426.4 | 89.3 | 232.6 | 47.6 | 709.3 | 7.13 |
| CompressCoT | 10.0 | 623.1 | 6.7 | 1673.7 | 52.5 | 1356.1 | 75.0 | 445.8 | 88.2 | 207.7 | 46.5 | 861.3 | 5.73 |
| CODI | 13.3 | 2798.7 | 6.7 | 613.5 | 50.0 | 518.6 | 72.4 | 537.5 | 87.2 | 238.1 | 45.9 | 941.3 | 5.18 |
| *SynAdapt* ($\tau$=1.0) | 13.3 | 718.8 | 16.7 | 620.7 | 57.5 | 591.9 | 75.6 | 739.4 | 88.5 | 253.5 | **50.3** | 584.9 | **9.14** |
| - Synthetic CCoT | 10.0 | 1743.9 | 16.7 | 475.9 | 52.5 | 510.2 | 73.2 | 599.8 | 87.8 | 266.7 | 48.0 | 719.3 | 7.10 |
| - Iterative Refine | 6.7 | 767.6 | 10.0 | 700.2 | 50.0 | 1073.9 | 76.0 | 993.8 | 85.4 | 728.9 | 45.6 | 852.9 | 5.68 |

Table 1: Comparison between our *SynAdapt* and efficient reasoning baselines for both Accuracy-Sensitive Scenario and Efficiency-Sensitive Scenario. **For the accuracy-sensitive scenario**, we set the threshold $\tau = 0.5$ for our method, meaning that questions with a difficulty score greater than 0.5 are routed to re-thinking, while others directly generate an answer based on the CCoT. **For the efficiency-sensitive scenario**, we set $\tau = 1.0$, meaning all questions are answered directly using the CCoT to achieve high efficiency. **Bold** and underlined numbers represent the best and second-best average accuracy, generation length and Rel-G score for each scenario.

accuracy while maintaining the shortest average generation length. CoT-FT fine-tunes directly on the full DCoT, improving accuracy on hard questions but also increasing generation length. TokenSkip selects parts of DCoT for fine-tuning, resulting in inconsistent CoT and performance degradation. NoThinking can skip CoT for reducing length, but often causes accuracy drops. CoD condenses each CoT step but cannot skip the unnecessary CoT in simple questions, resulting in a suboptimal accuracy-efficiency trade-off. TokenBudget dynamically allocates more tokens to harder questions, preserving accuracy but not reducing generation length effectively. In contrast, our method identifies hard questions and dynamically re-thinks them while directly generating answers for simple ones. It maintains similar accuracy compared to the raw model while reducing generation length, achieving the highest Rel-G score of 1.55 in the accuracy-sensitive scenario.

For the **efficiency-sensitive scenario**, our method with $\tau = 1.0$ significantly reduces the average generation length to just 584.9 tokens, while maintaining competitive accuracy compared to other baselines, as shown in the bottom part of Table 1. NoCoT-FT, which fine-tunes only on answers without CoT, leads to the accuracy drop. SelfTraining allows the LLM to search for the shortest correct CoT via best-of-$n$ sampling. But it struggles with harder questions and also results in a substantial drop in accuracy.

The three CCoT-based methods, Coconut, CompressCoT, and CODI, attempt to replace DCoT with CCoT. However, these methods only use a portion of DCoT or the last token as the alignment target when fine-tuning the LLM to learn CCoT. Due to the limited alignment signals, especially for hard questions, they achieve unsatisfactory accuracy. In contrast, our method introduces a more effective alignment target, the synthetic CCoT. By fully leveraging the alignment information from it, we enable more effective fine-tuning. Consequently, our method achieves the highest accuracy and the second shortest generation length in average, yielding the best trade-off with a Rel-G score of 9.14. We also present a representative case study in Figure 4 of Appendix.

Moreover, we evaluate our method under various $\tau$ values. As shown in Figure 3(a), our method consistently outperforms all other baselines, achieving the best accuracy-efficiency trade-off. As shown in the bottom of Table 1, we observe a significant performance decline when either Synthetic CCoT or Iterative Refinement is removed, which further highlights the importance of both components.

## 4.2 EVALUATION OF DIFFICULTY CLASSIFIER PERFORMANCE

**Experimental Settings** To evaluate the performance of our difficulty classifier, we use the **MATH500** dataset, treating questions with a difficulty level of 5 as hard and the rest as easy.

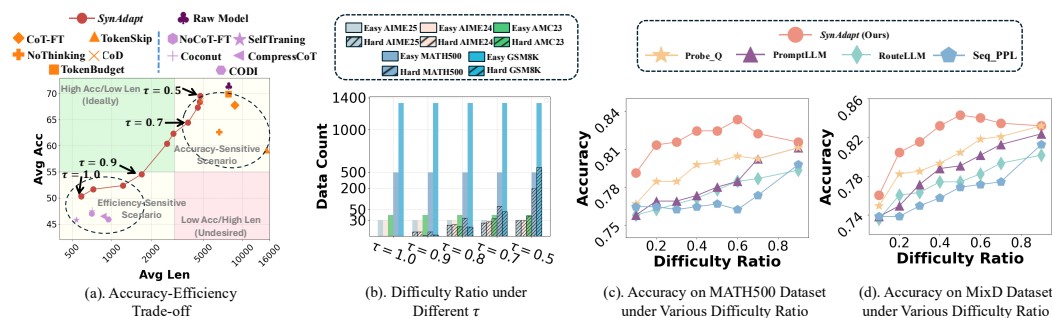

(a). Accuracy-Efficiency Trade-off  (b). Difficulty Ratio under Different $\tau$  (c). Accuracy on MATH500 Dataset under Various Difficulty Ratio  (d). Accuracy on MixD Dataset under Various Difficulty Ratio

Figure 3: (a) Accuracy-efficiency trade-off comparison between our method and other efficient reasoning baselines. (b). Difficulty ratio (The ratio of hard questions) of our method under different $\tau$ values across five benchmarks. (c/d). Accuracy under various difficulty ratios using different hard question identification methods on the MATH500 and MixD Datasets.

Additionally, we construct the **MixD** dataset by combining AIME25/AIME24/AMC23 and part of GSM8K. Questions from AIME25/AIME24/AMC23 are considered hard, while those from GSM8K are regarded as easy. We report macro precision (**Pre**), macro recall (**Rec**), and macro F1 (**F1**) of the hard question identification. We also report the accuracy of our method using different identification approaches, maintaining the same ratio of hard questions.

**Compared Methods** To demonstrate the effectiveness of our difficulty classifier, we consider several baselines for comparison: **Seq_PPL** (Mahaut et al., 2024) computes the PPL score for each question, treating those with high PPL as hard and others as easy. **PromptLLM** (Han et al., 2024) directly prompts the LLM to assess question difficulty. **RouteLLM** (Ong et al., 2024) trains an additional BERT model to judge question difficulty. We directly use their released weights. **Probe_Q** (Azaria & Mitchell, 2023) trains a simple classifier, consisting of two MLP layers, to assess difficulty based solely on the question. More details about the used datasets and the compared baselines are present in Appendix D.1 and D.2, respectively.

**Main Results** As shown in Table 2, our method, which identifies hard questions using both the question and CCoT, outperforms other baselines on both MATH500 and MixD datasets. Seq_PPL relies solely on the PPL score, which does not strongly correlate with question difficulty. PromptLLM prompts the LLM to assess difficulty, but this approach is unreliable due to the model's limitations in identifying hard questions. RouteLLM trains an additional BERT-based classifier, which incurs extra costs and struggles to effectively identify complex math questions re-

| Methods | MATH500 | | | MixD | | |
|---|---|---|---|---|---|---|
| | Pre | Rec | F1 | Pre | Rec | F1 |
| Seq_PPL | 37.46 | 35.27 | 36.10 | 28.93 | 23.70 | 25.51 |
| PromptLLM | 45.83 | 47.01 | 45.86 | 49.95 | 49.94 | 48.47 |
| RouteLLM | 46.03 | 47.27 | 31.21 | 13.37 | 48.00 | 20.91 |
| Probe_Q | 73.24 | 58.75 | 58.90 | **70.95** | 74.66 | 63.81 |
| *SynAdpat* | **79.47** | **62.42** | **63.11** | 62.71 | **81.02** | **78.32** |

Table 2: Comparison of *SynAdapt* and those baselines for hard question identification on MATH500 and MixD Datasets. **Bold** and underlined numbers indicate the best and second-best results, respectively.

quiring reasoning. Probe_Q trains a classifier based only on the question, which can identify explicit hard questions but misses those that look simple but actually hard. In contrast, our method can effectively identify those hard questions by using the reasoning information in corresponding CCoT. As shown in Figure 3 (b), it accurately identifies most difficult questions, such as those in AIME25/24, and AMC23.

Moreover, we also report the impact of different identification methods on overall performance in Figure 3 (c/d). We evaluate the problem-solving accuracy when using these methods under different difficulty ratios. As shown in Figure 3 (c/d), at the same difficulty ratio, our method can more accurately identify hard questions, route them for re-thinking, and achieve the best accuracy on both the MATH500 and MixD datasets. However, we observe a decrease in accuracy when the difficulty ratio exceeds 0.6. This is because easy questions are also routed for re-thinking, and excessive reasoning for simple questions will confuse the model, leading to incorrect answers.

## 4.3 ANALYSIS OF TRAINING EFFICIENCY

To evaluate training efficiency, we report the training cost of our method and other CCoT-based methods. As shown in Table 3, our method offers comparable efficiency to the baselines. While *SynAdapt* introduces additional synthetic CCoT generation, this process is highly efficient, accounting for **only 9.89%** of the total training cost. **Single CCoT generation only requires 10 seconds**, which is very fast.

| Modules | Time (min) | Percentage |
|---|---|---|
| Coconut | 740 | - |
| CompressCoT | 1192 | - |
| CODI | 1156 | - |
| *SynAdapt* | 1021 | 100% |
| LLM Training | 920 | 90.11% |
| **Synthetic CCoT Generation** (bs=16) | **101** | **9.89%** |
| ⇒ Single Synthetic CCoT Generation | 10s | 0.02% |

Table 3: Training time costs for different CCoT-based methods. We use a batch size (bs) of 16 during synthetic CCoT generation.

CompressCoT and CODI require autoregressive generation of CCoT during fine-tuning, leading to high training costs and low efficiency. Coconut gradually internalizes DCoT, and since the initial CCoT length is small, the training cost is relatively low. However, in the later stages, the cost still increases due to autoregressive generation. In contrast, *SynAdapt* **iteratively refines a draft CCoT rather than generating it autoregressively, effectively improving efficiency**. Therefore, our method achieves high training efficiency, demonstrating its practicality.

## 4.4 GENERALIZATION EVALUATION AND HYPERPARMETER ANALYSIS

To further demonstrate the generalization ability of *SynAdapt*, we evaluate it on **more domains**, including scientific question answering (GPQA-Diamond (Rein et al., 2024)) and code generation (HumanEval (Chen et al., 2021) and LiveCodeBench (Naman Jain, 2024)). As shown in Table 4, *SynAdapt* also exhibits superior performance in both scientific QA and coding tasks. With $\tau = 0.5$ for identifying hard questions requiring rethinking, our method achieves performance comparable to the raw model while reducing generation length. And with $\tau = 1.0$, which means no rethinking of any questions, *SynAdapt* still outperforms all other CCoT-based baselines. More results and analyses of our method on LiveCodeBench are provided in Appendix F. We also evaluate our method on **more LLM backbones**, such as DeepSeek-R1-Distill-Llama-8B and DeepSeek-R1-Distill-Qwen-1.5B (Guo et al., 2025). As shown in Table 5, *SynAdapt* consistently demonstrates superior performance under both $\tau = 0.5$ and $\tau = 1.0$ settings.

We conduct the **hyperparameter analysis** about CCoT length $m$ and refining iterations $k$. As shown in Figures 7 and 8 in the Appendix, our method remains effective and robust across various hyperparameter settings. Due to page limitations, more analyses and results are in Appendix G.

| Methods | GPQA-Diamond | | | HumanEval | | |
|---|---|---|---|---|---|---|
| | Acc ↑ | Len ↓ | Rel-G ↑ | Pass@1 ↑ | Len ↓ | Rel-G ↑ |
| Raw Model | **47.9** | 7847.1 | 1.00 | **75.6** | 4366.5 | 1.00 |
| *SynAdapt(τ=0.5)* | 47.5 | **6047.0** | **1.28** | 73.2 | **3503.6** | **1.21** |
| Coconut | **42.9** | 1406.6 | 4.99 | 70.7 | 750.6 | 5.44 |
| CompressCoT | 41.4 | 782.9 | 8.65 | 71.2 | 1386.5 | 2.97 |
| CODI | 40.9 | 676.6 | 9.89 | 65.9 | **602.6** | 6.32 |
| *SynAdapt(τ=1.0)* | 42.4 | **660.2** | **10.51** | **72.0** | 622.4 | **6.68** |

| Methods | R1-Llama-8B | | | R1-Qwen-1.5B | | |
|---|---|---|---|---|---|---|
| | Acc ↑ | Len ↓ | Rel-G ↑ | Acc ↑ | Len ↓ | Rel-G ↑ |
| Raw Model | **67.2** | 7998.4 | 1.00 | **57.6** | 9166.2 | 1.00 |
| *SynAdapt(τ=0.5)* | 66.1 | **6406.2** | **1.23** | 57.3 | **8836.5** | **1.03** |
| Coconut | 45.5 | 572.6 | 9.46 | 39.6 | 1767.1 | 3.57 |
| CompressCoT | 44.6 | 1834.3 | 2.89 | 38.2 | 1166.0 | 5.21 |
| CODI | 38.3 | **488.2** | 9.34 | 40.1 | 1566.5 | 4.07 |
| *SynAdapt(τ=1.0)* | **48.0** | 582.7 | **9.80** | **42.1** | 690.8 | **9.70** |

Table 4: Evaluation of our method across more domains, including GPQA-Diamond (Rein et al., 2024) for scientific question answering and HumanEval (Chen et al., 2021) for code generation.

Table 5: Evaluation of our method on DeepSeek-R1-Distill-Llama-8B and DeepSeek-R1-Distill-Qwen-1.5B backbones. We report the the average results across all five math benchmarks.

## 5 CONCLUSION

We propose a novel and efficient reasoning framework, *SynAdapt*, designed to help LLMs learn continuous CoT (CCoT). Before fine-tuning, we generate the synthetic CCoT, which serves as a more effective alignment target for learning CCoT. Additionally, we train a difficulty classifier that identifies hard questions by considering both the question and its corresponding CCoT. By dynamically prompting the LLM to re-think hard questions, our method can adapt to both accuracy-sensitive and efficiency-sensitive scenarios. Extensive experimental results across various benchmarks, domains and LLM backbones consistently demonstrate the effectiveness of *SynAdapt* for efficient reasoning.

## REPRODUCIBILITY STATEMENT

We provide the processing details of our training datasets and evaluation benchmarks in Sections C.1 and D.1. The statistic of these dataset are shown in Table 6 and 7. The implementation details of *SynAdapt* are provided in Section C.3 to facilitate reproducibility. We report the full settings used during LLM training and evaluation, including all hyperparameters. In addition, we have conducted experiment to analyze the impact of the hyperparameters in Section G and explain why we choose these setting.

To further facilitate reproducibility, we release all source code and the datasets used in our experiments in the supplementary materials. An anonymous repository containing the code and datasets is also provided for easy access by reviewers: https://anonymous.4open.science/r/SynAdapt_Review-E677. The repository includes a detailed user guide in the README files, covering installation, dependencies, and usage instructions.

## ETHICS STATEMENT

We adhere to the ICLR Code of Ethics in all aspects of this work. Our research utilizes exclusively publicly available repositories and datasets, ensuring full transparency and reproducibility. To rigorously validate the effectiveness of our proposed method and minimize the impact of randomness, we conduct extensive evaluations across a diverse range of domain tasks, large language model (LLM) backbones, and hyperparameter combinations. We not only evaluate the inference performance of our method, but also consider its training efficiency as critical factors in our analysis. To ensure a fair and comprehensive comparison, we rigorously assess training efficiency under consistent experimental conditions. All experiments are designed and reported in accordance with principles of responsible research, and we have conscientiously considered potential societal impacts in our work.

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

## THE USE OF LARGE LANGUAGE MODELS

In this paper, we strictly adhere to the usage policies of large language models (LLMs). LLMs were employed solely to assist with language polishing and to improve the readability of the manuscript. All generated content was carefully reviewed and verified by the authors before inclusion in the paper to ensure accuracy and integrity. No LLM outputs were used in a manner that could compromise reproducibility, scientific validity, or compliance with ethical standards.

## A  APPENDIX OVERVIEW

The appendix is organized into two main parts: Appendices B–D provide detailed related works and more experimental setup of our *SynAdapt*. Appendices E–G present additional case studies and experimental results, further demonstrating the effectiveness of our *SynAdapt*.

## B  DETAILS OF RELATED WORK

In this section, we provide a detailed overview of related works on LLM efficient reasoning, which can be broadly categorized into four main types: **SFT-based methods**, **RL-based methods**, **Prompt-based methods**, and **CCoT-based methods**.

For **SFT-based methods**, Yu et al. (2024) proposes to collect CoT and answer data from reasoning LLMs and directly discard the CoT part. And then they fine-tune the LLM using only the answers to help the model reduce reasoning length. Ma et al. (2025b) fine-tunes the LLM simultaneously on data with CoT and data without CoT, using specific instructions to distinguish between the two. During inference, they use the instructions to prevent the model from outputting CoT. Munkhbat et al. (2025) applies best-of-n sampling to LLM, selecting the shortest CoT, and fine-tune the model on these short CoTs to reduce reasoning length. Xia et al. (2025) assesses the semantic importance of tokens in the initial CoT, retaining only the most important tokens for fine-tuning the LLM. Kang et al. (2025) dynamically samples simplified CoTs from the model after each fine-tuning epoch for the next round of fine-tuning.

All of the above methods either discard the CoT or use a simplified version for fine-tuning the LLM to reduce reasoning length. While these approaches effectively shorten the reasoning length, they overlook important details in the original CoT, leading to significant performance degradation during further fine-tuning.

For **RL-based methods**, Arora & Zanette (2025) introduces a length-based reward, where shorter correct answers receive higher rewards, and uses policy gradient (PG) methods to fine-tune the LLM to reduce reasoning length. Luo et al. (2025) enhances this reward by comparing the generated answer length to a reference answer and applies PPO optimization for LLM fine-tuning. Yeo et al. (2025) further introduces a cosine-based reward and applies a penalty for exceeding the length limit. Aggarwal & Welleck (2025) uses length-constrained prompts to sampling data during RL fine-tuning. Shen et al. (2025a) employs SimPO to fine-tune the LLM using a length-preference dataset.

Although these RL-based methods can reduce reasoning length to some extent while maintaining LLM performance, RL fine-tuning requires significant resources. For example, they need to repeatedly sample new data for updating the action of LLM. Moreover, the reduction in length is limited and cannot be applied to those efficiency-sensitive scenarios. For instance, in real-life medical QA scenarios, efficiency is critical. Diagnosis advice must be concise, enabling doctors and patients to quickly access key details and conclusions, especially in emergencies. Previous studies (Kaddour et al., 2023; Liu et al., 2024) have highlighted that overly long responses can lead to errors, such as confusing similar drug names or omitting critical contraindications.

For **Prompt-based methods**, Renze & Guven (2024) proposes to prompt the LLM to perform CoT reasoning while explicitly instructing it to be concise. Xu et al. (2025a) focuses on adding instructions in the prompt to condense each reasoning step and limit verbosity. Lee et al. (2025) explores various prompt types to reduce reasoning length, such as prompting to output only numbers or only use bullet points. Han et al. (2024) estimates a token budget for each question, allocating more tokens for harder questions, and instructing the LLM to stay within this budget during reasoning for efficiency.

Most prompt-based methods reduce reasoning length by adding additional length constraint instructions in the prompt. While this approach is low-cost, its impact on reducing length is limited. LLMs still tend to generate redundant reasoning CoTs, especially when faced with hard questions.

For **CCoT-based methods**, Hao et al. (2024) was the first to propose to fine-tune the LLM to reason continuously and utilize the last hidden state as the continuous CoT (CCoT) to replace traditional discrete CoT (DCoT), which often contain redundant tokens. They introduce curriculum learning to gradually replace DCoT with CCoT during fine-tuning, without explicit alignment with the original DCoT. Xu et al. (2025b) is similar to Coconut, but it incorporates an additional assistant LLM with a projection module to generate the CCoT. Although it provides slight improvements, it also incurs additional resource costs. Shen et al. (2025b) employs self-distillation to learn CCoT by simultaneously fine-tuning on both DCoT and CCoT and explicitly aligns the last token hidden state between the two. Cheng & Van Durme (2024) measures token importance in advance and aligns the CCoT only with the hidden states of those important tokens in the DCoT.

Current CCoT-based methods can successfully compress reasoning steps into a latent space, replacing the original DCoT with a more efficient CCoT and significantly reducing generation length. However, they often suffer from unsatisfactory performance degradation. This is mainly because they either do not apply explicit alignment between DCoT and CCoT or only use partial DCoT (e.g., the last token or a subset of important tokens) to supervise CCoT learning. These weak supervisory signals fail to help LLM to learn a well CCoT representation, leading to significant performance drops. Therefore, designing stronger supervisory signals for CCoT learning is crucial for real-world applications.

## C    Details of Accuracy-Efficiency Trade-off Evaluation

### C.1    Dataset Details

**For the training set**, we use the DeepMath-103K dataset He et al. (2025), which contains numerous math problems with three distinct reasoning paths from DeepSeek-R1 Guo et al. (2025), covering various math topics and difficulty levels. For each question, we randomly select one reasoning path as the discrete CoT and exclude samples with reasoning paths exceeding 12,000 tokens. Moreover, as pointed out by Dong et al. (2024), the public datasets, containing numerous samples, suffer from a 'data contamination' issue, where some samples may be similar to evaluation benchmark. Directly training on this data may cause the model to memorize these samples, leading to unnaturally high performance. Additionally, including too many training samples introduces excessive training costs, which contradicts our goal of high efficiency. Therefore, we only sample a portion of the original DeepMath-103K dataset for training. Specifically, we randomly sample 10% of the training samples for each difficulty level to create the final **DeepMath** dataset, ensuring the distribution of question difficulty remains consistent. The total size of the DeepMath dataset is 9,660.

**For the test set**, we consider several widely adopted math-related benchmarks: **AIME25** mathai. (2024), **AIME24** Maxwell-Jia. (2024), **AMC23** zwhe99. (2024), **MATH500** Lightman et al. (2023), and **GSM8K** Cobbe et al. (2021). The difficulty of these benchmarks gradually decreases, covering a wide range from complex math competitions to simple grade school math. The details of both the train and test dataset sizes are shown in Table 6.

| Train Dataset | Test Dataset | | | | |
|---|---|---|---|---|---|
| **DeepMath** | **AIME25** | **AIME24** | **AMC23** | **MATH500** | **GSM8K** |
| 9660 | 30 | 30 | 40 | 500 | 1319 |

Table 6: The size of our used train dataset and five math-related evaluation benchmarks, covering various difficulty levels.

### C.2    Baselines Details

Here, we provide more details about all the compared efficient reasoning baselines. We consider not only CCoT-based baselines but also other SFT-based and prompt-based methods. We exclude

RL-based methods, as these require substantial resources to apply RL learning to LLMs, making them inefficient and impractical for real-world applications.

We further mainly categorize these baselines into two scenarios based on their focus. Baselines for the **accuracy-sensitive scenario** primarily aim to maintain performance while shortening the generation length. Here are the details of these baselines:

**CoT-FT** belongs to SFT-based methods. We directly uses the CoT and answers from the training set, to supervise fine-tune (SFT) the LLM. This method aims to maintain accuracy while slightly reducing the generation length.

**TokenSkip** (Xia et al., 2025) belongs to SFT-based methods. As proposed by TokenSkip, different tokens in the CoT have varying semantic importance, and tokens with low semantic value can be skipped during SFT of the LLM. Specifically, we use LLMLingua-2 (Pan et al., 2024) to assess the importance of each token and obtain a compressed CoT. We set the compression ratio to 0.7 because too low ratio will make the CoT inconsistent for fine-tuning while too low ratio only provides a slight reduction in generation length. We utilize the compressed CoT along with corresponding answer to fine-tune LLM to reduce generation length while maintaining performance.

**NoThinking** (Ma et al., 2025a) is a prompt-based method. NoThinking proposes to directly prompt the LLM to avoid generating a CoT, which effectively reduces the generation length with fine-tuning process. Specifically, we append the instruction "*Okay, I think I have finished thinking.</think>*" to the initial prompt, instructing the LLM to skip reasoning and directly output the answer without CoT.

**CoD** (Xu et al., 2025a) is another prompt-based method. Different from NoThinking directly prompts LLM to skip reasoning and do not output CoT, Chain-of-Draft (CoD) preserves the reasoning process but condenses each reasoning step by inserting the "*only keep a minimum draft for each thinking step, with 5 words at most.*" instruction.

**TokenBudget** (Han et al., 2024) is also a prompt-based method. Following TokenBudget, we prompt the LLM in advance to estimate the difficulty of each question and determine the essential token budget. During inference, we incorporate the token budget into the initial prompt by adding the instruction, "*Let's think step by step and use fewer than [[Token Budget]] tokens*", guiding the LLM to reduce unnecessary generation.

In contrast, baselines for the **efficiency-sensitive scenario** prioritize improving efficiency, even at the cost of performance. Here are the details of these baselines:

**NoCoT-FT** (Yu et al., 2024) is an SFT-based method. However, unlike previous SFT-based methods, NoCoT-FT distills the ability from the reasoning model to the model that does not output any CoT, by fine-tuning solely on the answer part from the reasoning model. Specifically, we discard the CoT part in our training set and fine-tune the LLM only with the answer.

**SelfTraining** (Munkhbat et al., 2025) is another SFT-based method. As proposed by SelfTraining, we apply best-of-n sampling to the LLM to generate multiple answers for each question, then select the shortest correct answer to fine-tune the LLM and reduce generation length. During sampling, we also provide demonstrations as few-shots to instruct the LLM to generate the answer directly without CoT. The sampled answers are then used to fine-tune the LLM to skip the CoT.

**Coconut** (Hao et al., 2024) is one of CCoT-based methods. According to Coconut, we apply curriculum learning to help the LLM gradually learn Continous CoT (CCoT). Specifically, we fine-tune the LLM for 3 epochs, gradually reducing the initial DCoT tokens to none as the epochs progress, and replacing them with CCoT. Finally, we can internalize the DCoT into the CCoT.

**CompressCoT** (Cheng & Van Durme, 2024) belongs to CCoT-based methods Following Compress-CoT, we first identify important tokens in the discrete CoT using LLMLingua-2 (Pan et al., 2024) and compute the mid-layer hidden states of these tokens as the target. We then fine-tune the LLM with the LoRA module to generate the CCoT similar to target. Simultaneously, we fine-tune another LoRA module to predict the correct answer based on the CCoT. During inference, we first use the prior LoRA module to generate the CCoT and then use the other LoRA module to generate the answer based on it.

**CODI** (Shen et al., 2025b) is another CCoT-based methods. As proposed by CODI, we fine-tune the LLM with two tasks: the teacher task, which generates the discrete CoT tokens and the final correct

answer, and the student task, which generates the CCoT and the correct answer. We then explicitly align the last token hidden states from the DCoT and CCoT to achieve self-distillation from DCoT to CCoT.

### C.3 IMPLEMENTATION DETAILS OF OUR *SynAdapt*

We adopt the DeepSeek-R1-Distill-Qwen-7B (Guo et al., 2025) as the LLM backbone and we also evaluate the our method on other backbones in Section 4.4. For the **Synthetic CCoT generation**, we fix the LLM backbone and make the randomly initialized synthetic CCoT to be trainable. The length of synthetic CCoT is set as $m = 512$ and the we optimize it using the learning rate at 1e-3 for 32 steps. During optimization, we use a batch size of 16 to ensure high efficiency.

For **Synthetic CCoT Enhanced Fine-tuning**, we use LoRA (Hu et al., 2022) to fine-tune the LLM for learning CCoT. The lora rank is set to be 8 and the alpha value at 32. We use the Deepspeed (Rasley et al., 2020) framework to fine-tune the LLM. We fine-tine LLM for 3 epochs with a batch size of 1 and a gradient accumulation step of 16. We employ the AdamW optimizer with a learning rate set to 4e-5. The refinement steps of the draft is $k = 4$ and the length of CCoT is also $m = 512$. We also analyze these hyperparameters in Section 4.4.

For **Adaptive Reasoning via CCoT**, we firstly generate the CCoT with the length of 512 and use the difficulty classifier to judge the difficulty score $\tau$ based on question and CCoT. The score ranges from 0 to 1, with scores below the threshold $\tau$ considered as simple, and those above as hard. For the efficiency-sensitive scenario, we set $\tau = 1.0$, treating all questions as simple. For the accuracy-sensitive scenario, we set $\tau = 0.5$ to classify some questions as hard. We also try more $\tau$ values, as shown in Figure 3(a). During answer generation, we use greedy decoding and set the maximum generation length to 32,768 tokens. The generation prompt and the prompt for re-thinking hard questions are provided in Appendix H. All our training and evaluation experiments are conducted on the H20 GPU.

## D DETAILS OF DIFFICULTY CLASSIFIER EVALUATION

### D.1 DATASET DETAILS

Here, we will introduce the details of the two datasets used to evaluate the hard question identification performance. For the **MATH500** dataset (Lightman et al., 2023), we use the original difficulty labels, which range from 1 to 5, with higher values indicating more difficult questions. Questions with a difficulty level of 5 are considered hard, while the others are easy. The detailed statistics are shown in Table 7.

For the **MixD** dataset, we combine questions from AIME25, AIME24, and AMC23 to form the hard question set. Questions from the GSM8K dataset are considered easy. We random select 20% questions from GSM8K randomly selected to form the easy question set to avoid severe data imbalance problem. We then mix both the hard and easy questions to create our MixD dataset. The detailed statistics are shown in Table 7.

| Dataset | Total Cnt | Number of Hard | Number of Easy |
|---------|-----------|----------------|----------------|
| **MATH500** | 500 | 134 | 366 |
| **MixD** | 363 | 100 | 263 |

Table 7: The statistics for our used test dataset used to evaluate hard question identification performance.

### D.2 BASELINES DETAILS

In this section, we provide a more detailed introduction to the baselines for hard question identification as follows:

**Seq_PPL** (Mahaut et al., 2024) uses sequence probability (PPL) to reflect the confidence of the LLM. We compute the PPL of the LLM on each question, which is equivalent to the sequence probability

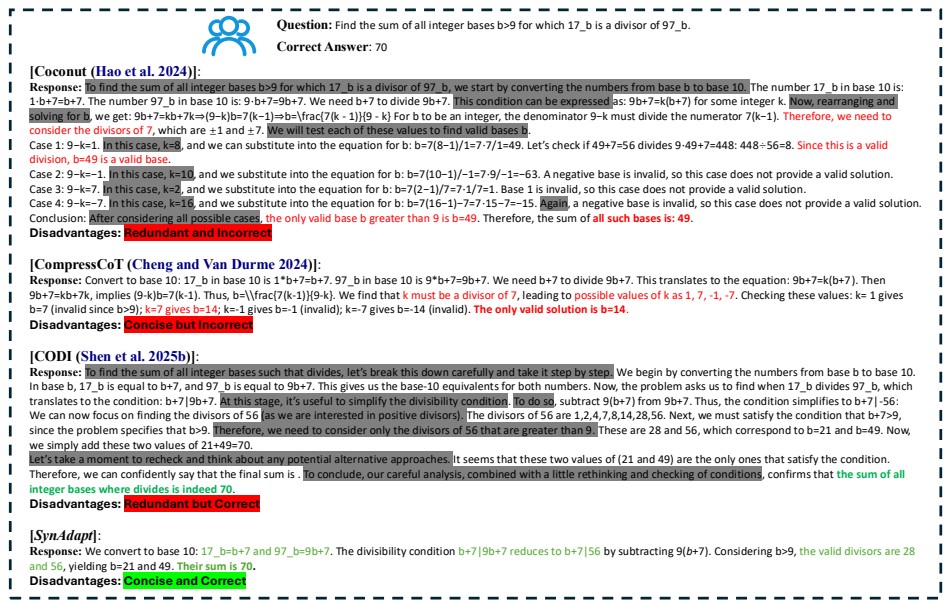

Figure 4: An representative example of the generated output from our *SynAdapt* and other CCoT-based baselines is provided. We highlight the crucial wrong steps that lead to incorrect answers in red, and the correct reasoning steps in green. Redundant parts in the answer are marked with a gray background. We also provide short analyses explaining the disadvantages or advantages of the generated responses.

of the question. We treat those questions with high PPL are considered hard, while those with low PPL are categorized as simple.

**PromptLLM** (Han et al., 2024) prompts the LLM to assess the difficulty of a question and predict the essential token budget required for solving it. We also prompt the LLM to predict the token budget and restrict the range to 128-32,768 tokens. Questions that require a high token budget are considered hard, while those with a low token budget are classified as simple.

**RouteLLM** (Ong et al., 2024) trains a hard question classifier using a BERT backbone. The classifier assigns high scores to hard questions and routes them to stronger LLMs, such as GPT-4 OpenAI (2025), while easier questions are processed by weaker LLMs, like Mixtral-8x7B Jiang et al. (2024). Therefore, we directly use their released model weights [2] and classify those questions with high scores as hard.

**Probe_Q** (Azaria & Mitchell, 2023) trains a classifier based on the LLM's hidden state to assess truthfulness. Similarly, we provide the LLM with the question and train a classifier to evaluate difficulty based on the last token's hidden state from LLM. This approach is similar to ours, but it does not leverage information from the CCoT for assessing question difficulty.

# E    CASE STUDIES

## E.1    RESPONSE EXAMPLE FROM VARIOUS BASELINES

We provide a representative example to demonstrate the effectiveness of our *SynAdapt* by comparing its generated response with those from other CCoT-based baselines, including Coconut, CompressCoT, and CODI.

As shown in Figure 4, the response from **Coconut** contains numerous redundant parts, which primarily serve communication or linguistic purposes, rather than contributing to the reasoning process needed to derive the correct answer. Moreover, the answer generated is incorrect, highlighting that indirect

---

[2]https://huggingface.co/routellm/bert_gpt4_augmented

training without explicit alignment with DCoT fails to effectively learn CCoT. **CompressCoT** successfully generates a concise response without redundancy but still outputs the wrong answer. This is because it aligns only with a subset of isolated, incoherent DCoT tokens, which fail to capture the full reasoning process, resulting in performance degradation. For **CODI**, the generated response provides the correct answer but retains redundant parts. This occurs because it applies alignment only at the final position, limiting its ability to learn CCoT and produce concise output.

In contrast, our method generates both a concise and correct answer. This is due to our use of synthetic CCoT as the fine alignment target and applying full alignment during CCoT fine-tuning. These results strongly demonstrate the effectiveness of our method for efficient reasoning.

### E.2 CCoT for Hard Question Example

We provide an illustrative example demonstrating that solely relying on CCoT is insufficient to solve hard questions. As shown in Figure 6, when the LLM relies only on CCoT, it generates a concise but incorrect answer. It may be because CCoT restricts the LLM's ability to verify reasoning steps, confining it to the incorrect answer. However, when prompted to re-think the question, the LLM can rectify the previous mistake and derive the right answer. This effectively demonstrates that compressing DCoT into CCoT inevitably results in information loss, limiting the model's reflective ability and leading to incorrect answers.

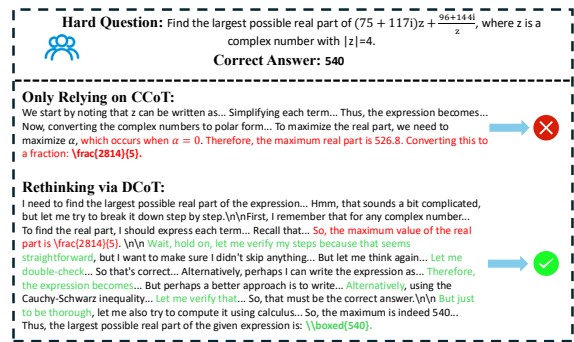

Figure 5: An illustrative example of solving hard question relying solely on CCoT or rethinking via DCoT. We highlight the crucial wrong steps that lead to incorrect answers in red, and the correct reasoning steps in green.

### E.3 Indistinguishable Hard Question Example

We provide an illustrative example showing how some hard questions are similar to simple ones, making them difficult to distinguish. As seen in Figure 6, both the easy and hard questions are very similar, both focusing on the quaternions topic and are short in length. If we only assess difficulty based on the question itself, both would be categorized as easy, leading to performance degradation.

However, when considering the CoT process, there exist significant differences. For the easy question, the CoT is short and easily leads to the correct answer. In contrast, the hard question involves more reasoning steps and a longer CoT. By incorporating both the CoT and the question, we can accurately identify these indistinguishable

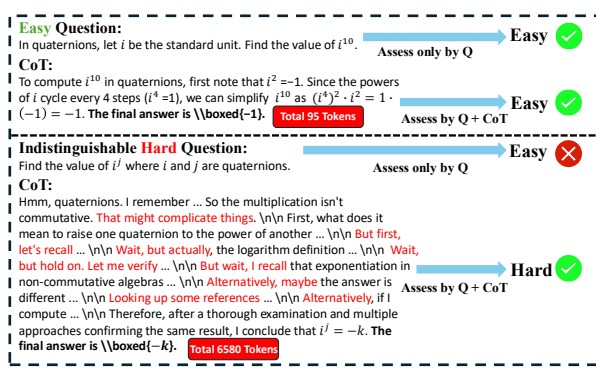

Figure 6: An illustrative example of an easy question and the similar hard question, with their corresponding CoT processes. We also present the identification results using only the question or both the question and CoT. The key differences from the CoT of the hard question, compared to the easy question, are highlighted in red color.

hard questions. This highlights the value of reasoning information in identifying hard questions. This is also why our difficulty classifier is build up on both the question and CCoT, which can effectively utilize the information in CCoT.

## F More Evaluation of *SynAdapt*'s Generalization

To further assess the generalization ability of our *SynAdapt*, we extend the evaluation beyond the mathematical domain to more various domains, including GPQA-Diamond (Rein et al., 2024) for scientific question answering, HumanEval (Chen et al., 2021) and LiveCodeBench (Naman Jain, 2024) for code generation. The results on GPQA-Diamond and HumanEval are shown in Table 4, while those on LiveCodeBench are presented in Table 8.

| Methods | LiveCodeBench | | |
|---|---|---|---|
| | Pass@1 ↑ | Len ↓ | Rel-G ↑ |
| Raw Model | **46.4** | 8642.8 | 1.00 |
| *SynAdapt($\tau$=0.5)* | 41.1 | **6689.8** | **1.14** |
| Coconut | 26.6 | 900.3 | 5.50 |
| CompressCoT | 26.4 | 1323.4 | 3.72 |
| CODI | 25.4 | 689.0 | 6.87 |
| *SynAdapt($\tau$=1.0)* | **26.7** | **658.5** | **7.55** |

Table 8: Evaluation results of our method and those CCoT-based methods on Live-CodeBench Naman Jain (2024) for code generation task.

As shown in Table 8, *SynAdapt* remains effective for the **LiveCodeBench** benchmark. Under the accuracy-sensitive setting with $\tau = 0.5$, our method achieves comparable performance to the raw model while substantially reducing generation length, resulting in a Rel-G score of 1.14. Under the efficiency-sensitive setting with $\tau = 1.0$, it outperforms the other existing CCoT-based methods and achieves the best Rel-G score of 7.55. These results provide additional evidence that our method generalizes well to those coding tasks, rather than being limited to the mathematical domain.

## G Hyperparameter Analysis

**The Length of CCoT $m$.** We analyze the hyperparameter $m$ in our method, which controls the length of the CCoT. As shown in Figure 7(a), increasing $m$ leads to higher accuracy as well as longer generation length. This is mainly because a longer CCoT contains more reasoning steps, which benefits problem solving and improves accuracy. At the same time, longer CCoT also boosts the likelihood of model to generate redundant content, such as repeated verification steps, which simultaneously increases the generation length.

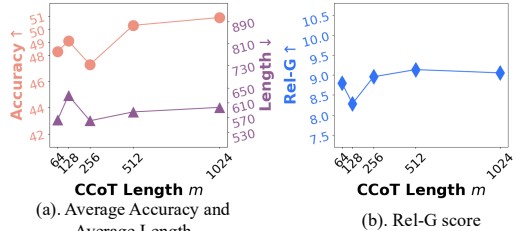

(a). Average Accuracy and Average Length

(b). Rel-G score

Figure 7: The performance of our methods when using different CCoT Length $m$. We report the average results across five math benchmarks.

To further measure the trade off performance between high accuracy and low generation length, we compute the Rel-G score to capture the actual performance gain, as shown in Equation 9. As illustrated in Figure 7(b), initially increasing the CCoT length $m$ improves accuracy and leads to a higher Rel-G score. However, further increasing $m$ causes the model to generate excessive redundant content, resulting in a decline in the Rel-G score. Overall, setting $m = 512$ yields the best Rel-G score, indicating the optimal balance between accuracy and efficiency.

**The Refining Iterations of CCoT $k$.** We also analyze the hyperparameter $k$ in our method, which controls the refining iterations of CCoT. As shown in Figure 8(a), in the initial stage, increasing $k$ allows the CCoTs to progressively refine potentially incorrect reasoning steps, enabling the LLM to produce more accurate and concise answers. As a result, accuracy increases while the generation length decreases. However, when $k$ is further increased beyond 4, redundant refinement steps may confuse the LLM, leading to longer generation lengths and a slight decrease in accuracy.

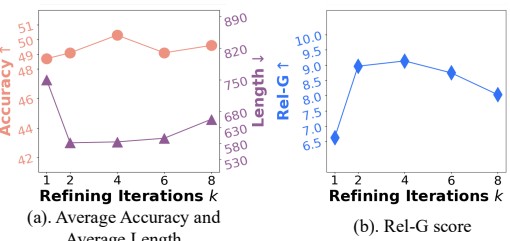

(a). Average Accuracy and Average Length

(b). Rel-G score

Figure 8: The performance of our methods when using different refining iterations $k$ for CCoT generation. We report the average results across five math benchmarks.

To further measure the accuracy-efficiency trade-off, we compute the Rel-G score, following Equation 9. As shown in Figure 8(b), initially increasing $k$ improves accuracy and reduces generation length, resulting in a higher Rel-G score. However, when $k$ exceeds 4, the increase in generation length and slight decrease in accuracy lead to a lower Rel-G score. Therefore, setting $k = 4$ achieves the best Rel-G score, indicating the optimal trade-off between accuracy and efficiency.

## H USED PROMPT TEMPLATES

In this section, we present the prompts used in our method. For easy questions, we directly prompt the LLM to generate an answer based on the CCoT, as shown in Figure 9. For hard questions, we prompt the LLM to re-think and generate discrete CoT, condensing each reasoning step, as illustrated in Figure 10.

```
< | begin_of_sentence | >
Please reason step by step, and put your final answer within \\boxed{{}}.

< | User | >
[[INSERT USER QUESTION HERE]]

< | Assistant | >
<think> [[INSERT CCoT HERE]] </think>
```

Figure 9: The prompt used for directly generating answers based on the CCoT.

```
< | begin_of_sentence | >
Think step by step, but only keep minimum draft for each thinking step,
with 5 words at most.
Return the answer at the end of the response within \\boxed{{}}.

< | User | >
[[INSERT USER QUESTION HERE]]

< | Assistant | >
<think>
```

Figure 10: The prompt used for re-thinking hard questions via discrete CoT process while condensing each CoT step.

