# OpenReview forum: "SynAdapt: Learning Adaptive Reasoning in Large Language Models via Synthetic Continuous Chain-of-Thought"
_ICLR.cc/2026/Conference — ICLR 2026 Conference Withdrawn Submission_

### Official Review · Reviewer_ELKj · 2025-10-28

**Soundness:** 3
**Presentation:** 3
**Contribution:** 3
**Rating:** 6
**Confidence:** 4

**Summary:**

The paper proposes an innovative efficient reasoning framework for adaptive reasoning in LLMs via synthetic continuous CoT. The synthetic CCoT explicitly guides the LLM to learn CCoT and derive accurate answers directly. Further, the paper proposes a classifier to distinguish easy and hard queries. Finally, extensive experiments are conducted to verify the usefulness of the proposed method.

**Strengths:**

1. Adaptive reasoning is of practical use due to the need to save resources.
2. The paper is very well written and easy to follow.
3. Experiments have clearly demonstrated the capability of the proposed model in balancing effectiveness and efficiency for inference.

**Weaknesses:**

1. The proposed method is relatively weak in effectiveness.
2. CCoT method lacks explainability, which deviates from explainable inference in daily use.
3. The classifier introduces an additional hyperparameter to control easy and hard queries, which is difficult to set.

**Questions:**

1. In my eyes, one of the major weakness in this paper is the deficiency in model accuracy. The strengths of CCoT lies in generating less tokens, which improves the efficiency. However, it harms the effectiveness, as shown in Table 1.
2. The classifier is used to predict easy and hard queries. However, how to set the threshold is difficult. In other words, how hard is hard and how easy is easy?
3. The generated synthetic CCoT is used to fine-tune LLM for better CCoT understanding, which is reasonable. But the generation of synthetic CCoT is separately optimized from fine-tuning. Hence, I am concerned about the problem that the generated $Z_{syn}$ could be in different subspace from $Z_{final}$, which leads to sub-optimal results.

---

> ### Author Response · Authors · 2025-11-17
> **Authors' Response (1/2)**
>
> We sincerely thank you for your feedback and suggestions! We address all your concerns below:
>
> > W1: The proposed method is relatively weak in effectiveness.
> Q1: One of the major weakness in this paper is the deficiency in model accuracy...
>
> Thank you for your question. Indeed, there may be some misunderstanding about our results in Table 1. First of all , as defined in Table 1, we consider two scenarios for efficient reasoning: (1) **Accuracy-sensitive scenario**, which requires maintaining accuracy while reducing generation length, and (2) **Efficiency-sensitive scenario**, which allows a certain accuracy drop in exchange for significant length reduction.
>
> Secondly, thanks to the adaptiveness of our method via threshold $\tau$, as shown in Section 3.2, **our approach can be applied to both scenarios**. As shown in Table 1, **in the accuracy-sensitive scenario**, **our method achieves accuracy comparable to the raw model** while substantially reducing generation length. For clarity, we restate the key results again below:
>
> || Average Acc $\Uparrow$ | Average Len $\Downarrow$ | Average  Rel-G $\Uparrow$ |
> |-|-|-|-|
> |Raw Model|**73.3**|7786.84|1.00|
> |**SynAdapt when $\tau=0.5$ (Ours)**|69.0 ($\downarrow$ 4.3)|**4694.80** ($\downarrow$ 3092.04)|**1.58** ($\uparrow$ 0.58)|
>
> As shown above, our method **reduces the average generation length by nearly 50%** (from 7786.84 to 4694.80) while **achieving comparable accuracy** (only 4.3 drop). Overall, the **Rel-G score of 1.58** demonstrates that our method provides a strong accuracy–efficiency trade-off in the accuracy-sensitive scenario.
>
>
> > W3:  The classifier introduces an additional hyperparameter to control easy and hard queries, which is difficult to set.
> Q2: How to set the threshold is difficult. In other words, how hard is hard and how easy is easy?
>
> Thanks for your interesting question. Firstly, we believe that **the hyperparameter $\tau$**, which determines the boundary between hard and easy questions, **should be determined according to different application scenarios**. Specifically, **for accuracy-sensitive scenarios**, **a lower threshold (e.g., $\tau$ = 0.5) is preferable**, as it treats more questions as hard and allocates more reasoning budget via DCoT, thereby maintaining accuracy comparable to the raw model. In contrast, **for efficiency-sensitive scenarios**, **a higher threshold (e.g., $\tau$ = 1.0)** allows most questions to be treated as easy and directly answered via CCoT, substantially reducing generation length. Therefore, we **believe the choice of $\tau$ should depend on scenario-specific requirements**. Moreover, the threshold $\tau$ allows our method to **be flexibly applied to both accuracy-sensitive and efficiency-sensitive scenarios** by simply adjusting its value. This flexibility comes with minimal additional cost, which the other baselines do not offer, as shown in Table 1.
>
> Furthermore, we conducted a comprehensive evaluation of **our method under different values of $\tau$**, as presented in **Figure 3(a) and Lines 370–373**. As shown in the figure, **our method consistently outperforms all baselines across different $\tau$ values**, which strongly supports its effectiveness in achieving a superior accuracy–efficiency trade-off for LLM reasoning.

---

> ### Author Response · Authors · 2025-11-17
> **Authors' Response (2/2)**
>
> > Q3: The generation of synthetic CCoT is separately optimized from fine-tuning. Hence, I am concerned about the problem that the generated synthetic CCoT $Z_{syn}$ could be in different subspace from target CCoT $Z_{final}$, which leads to sub-optimal results.
>
> Thanks for your question. We believe the concern arises from a misunderstanding about the nature of the synthetic CCoT $Z_{syn}$ and the target CCoT $Z_{final}$. **In fact, both CCoTs inherently lie in the same reasoning subspace defined by the raw LLM**. As shown in Figure 1, **when generating $Z_{syn}$, we freeze all parameters of the raw model** and only optimize the synthetic CCoT itself. Because the forward computation is fully determined by the fixed LLM, **the optimized $Z_{syn}$ is always constrained to the model’s intrinsic reasoning subspace**. Moreover, **the target CCoT $Z_{final}$ is also learned by the same raw model**, and therefore **naturally resides in exactly the same subspace**. Therefore, the premise that $Z_{syn}$ and $Z_{final}$ may lie in different subspaces does not hold.
>
> To further address the concern, we **additionally evaluate an end-to-end training strategy** that bypasses synthetic CCoT and directly fine-tunes the model to learn the target CCoT, which corresponds to the CODI baseline in Table 1.  Here we restate the key results in Table 1 for clarity:
>
> || Average Acc $\Uparrow$ | Average Len $\Downarrow$ | Average  Rel-G $\Uparrow$ |
> |-|-|-|-|
> |**Seperately Training (Ours)**|**50.3**|**584.9**|**9.14**|
> |End-to-End Training|45.9($\downarrow$ 4.4)|941.3($\uparrow$ 356.4)|5.18 ($\downarrow$ 3.96)|
>
> We mark out better results in **bold**.
> As shown in the above results, **our method achieves better accuracy and efficiency** compared with end-to-end training. This improvement primarily comes from using **the synthetic CCoT** as a high quality alignment target, which **remains naturally aligned with the reasoning subspace of the raw LLM**.
>
>
> ----
> We sincerely appreciate your time and effort in reviewing our paper.
> We have mark the key points of our response in **bold** for easier reading.
> **We look forward to your further feedback on our paper**. If you have any additional questions, we would be happy to discuss them!
> **If our reponses can address your concerns, can you consider to raise you score?** We will deeply appreciate for it!
> Once again, thank you very much for your thoughtful and constructive review.

---

> > ### Comment · Reviewer_ELKj · 2025-11-27
> >
> > Thank you for your further clarification. I do not think I have any misunderstanding on the first question. From my view point, the efficiency-sensitive scenario makes little sense because the effectiveness drops too much and all the methods cannot be used. For the accuracy-sensitive scenarios, the proposed method cannot achieve the best performance, but it can significantly reduce the number of generated tokens. The effectiveness is not the strength of the proposed method, while the low cost should be emphasized. I will keep my score.

---

> ### Author Response · Authors · 2025-11-28
> **Further Authors' Response for Reviewer ELKj (1/1)**
>
> Thank you very much again for your timely response and positive review of our paper. Here we want to add additional explanations about your remaining concern in your first question.
>
> > Q1：The efficiency-sensitive scenario makes little sense because the effectiveness drops too much and all the methods cannot be used. ... The effectiveness is not the strength of the proposed method, while the low cost should be emphasized.
>
> **We fully agree with your viewpoint** that efficiency-sensitive scenarios may be less practical than accuracy-sensitive scenarios in real-world applications due to their noticeable performance drop. However, **we want to emphasize that including the efficiency-sensitive scenario in our evaluation is still essential: it enables a more complete and fair comparison** between our method and existing baselines. In fact, **a number of prior works**, such as Coconut, CODI, and CompressCoT, are explicitly **designed for efficiency-sensitive settings**. Incorporating them ensures our evaluation to be as complete as possible, making our results more comprehensive and convincing.
>
> We also appreciate your suggestion about highlighting the low cost of our method. **This is very correct: our focus is on efficient reasoning rather than improving reasoning ability**. Moreover, to enable a fair comparison that **jointly considers effectiveness and efficiency**, we introduce **a new metric, *Rel-G***, which quantifies the relative efficiency gain per unit accuracy cost. As shown in Table 1, our method achieves the highest *Rel-G* score. **This is the key point we really want to emphasize: our approach attains the best trade-off between efficiency and effectiveness**. In other words, our method obtains substantial efficiency improvements at only a minimal cost in accuracy. This trend is also reflected in Figure 3(a) and the accompanying discussion.
>
> Anyway, we sincerely appreciate your constructive feedback once again. **We will incorporate stronger emphasis on the low-cost nature of our method in future version** to further enhance the clarity and persuasiveness of our contribution.
>
> ----
>
> Thanks for your efforts once again!
> **We look forward to your further feedback on our paper**. If you have any additional questions, we would be happy to discuss them at your earliest convenience !

---

### Official Review · Reviewer_pikB · 2025-10-29

**Soundness:** 2
**Presentation:** 3
**Contribution:** 2
**Rating:** 4
**Confidence:** 2

**Summary:**

This paper introduces SynAdapt, a framework for efficient and adaptive reasoning in LLMs by generating and aligning synthetic continuous Chain-of-Thought (CCoT) representations. The authors propose synthesizing a continuous CoT to serve as an explicit and optimized alignment target, rather than relying on discrete or partial alignments as in prior work. SynAdapt also integrates a difficulty classifier that uses both the input question and its CCoT to identify challenging problems, adaptively prompting the LLM to re-think harder cases using discrete, step-wise reasoning.

**Strengths:**

1. The paper brings a new perspective to CCoT learning in LLMs, addressing weaknesses of prior partial or indirect alignment approaches. Specifically, SynAdapt’s use of synthetic, explicitly optimized CCoTs as full alignment targets for fine-tuning is clearly articulated and represents a concrete methodological advance.
2. The inclusion of an adaptive, CCoT-informed difficulty classifier is well-motivated and shows robust empirical performance over alternatives that rely on perplexity, prompting, or question-only signals.

**Weaknesses:**

1. Missing Discussion of Key Related Recent Works. [1] also proposed a module to classify the questions based on the questions’ complexity.
[1] X. Chen, S. Zhou, K. Liang, and X. Liu, “Distilling reasoning ability from large language models with adaptive thinking,” IEEE Transactions on Neural Networks and Learning Systems, pp. 1–14, 2025.
2. Clarity and Interpretation of Mathematical Descriptions:
- the motivation for aligning only the eot token’s hidden states is given for overfitting prevention, but why this is preferable to multi-token or more structured alignments is left vague.
- The reason why iteratively refine is needed is not clearly stated in the text.
- The hidden dimensions such as Zsyn are not marked, which greatly affects the reader's understanding.
3. The authors did not conduct experiments on the sensitivity of the length of the synthetic CCoT, denoted as m. In my opinion, the value of m should have a significant impact on the performance of CCoT. Moreover, since m is set to a fixed value, I wonder whether this is appropriate for chain-of-thought reasoning, whose information content can vary greatly across different problems.
4. Generalization beyond mathematical reasoning is claimed but not demonstrated. The method is never tested on tasks outside math QA
5. Using token count or sequence length as an efficiency metric is unfair, since the proposed method includes several preprocessing steps—such as iterative refine and difficulty estimation—before generating the chain of thought. Therefore, inference time would be a fairer basis for comparison.

**Questions:**

See weaknesses

---

> ### Author Response · Authors · 2025-11-17
> **Authors' Response (1/4)**
>
> We sincerely thank you for your feedback and suggestions! We address all your concerns below:
>
> > W1: Missing Discussion of Key Related Recent Works.
>
> Thank you for your suggestion. Although the paper you cited [1] also focuses on efficient adaptive reasoning, there are some differences compared to our method. The main distinction is that **[1] employs a post-thinking approach**, prompting the LLM to directly output the answer and then generate the rationale (CoT) to verify correctness. **These rationales still consist of discrete tokens**, introducing efficiency costs due to autoregressive generation. **In contrast, our method uses CCoT to replace DCoT** and leverages iterative refinement rather than autoregressive generation, ensuring higher efficiency. Therefore, **[1] belongs to the DCoT paradigm, while our approach falls within the CCoT reasoning domain**.
>
> Moreover, **[1] judges question difficulty only based on the characteristics of the question itself**, using an attention mechanism to extract features. However, **our method enhances difficulty estimation by leveraging reasoning information** contained in the CCoT, achieving better performance than using only the question features, as shown in Table 2 and Section 4.2.
>
> To some extent, **our method is orthogonal to [1], and it is possible to combine the two approaches** to achieve even more efficient reasoning. For example, the post-thinking process in [1] could potentially utilize CCoT to further improve verification efficiency. We leave this exploration for future work.
> After all, **we promise to include this discussion of the related work [1]** in the camera-ready version to make our paper more complete.
>
> [1] X. Chen, S. Zhou, K. Liang, and X. Liu, “Distilling reasoning ability from large language models with adaptive thinking,” IEEE Transactions on Neural Networks and Learning Systems, pp. 1–14, 2025.
>
>
> > W2.1: the motivation for aligning only the eot token’s hidden states is given for overfitting prevention, but why this is preferable to multi-token or more structured alignments is left vague.
>
> Thank you for the interesting question. Indeed, we do not include more tokens from the DCoT (i.e., multi-token extraction) or more structured alignments when generating the synthetic CCoT because **selecting only a small subset of tokens from the DCoT introduces semantic inconsistency**. This issue is similar to the main limitation of CompressCoT, as illustrated in Figure 1 and Lines 67–71. In fact, **extracting a few isolated tokens** cannot adequately capture the full reasoning process encoded in the original DCoT and **will inevitably lead to performance degradation due to an inconsistent optimization target**.
>
> Moreover, we conducted additional experiments **comparing our method with distilling multiple tokens from the DCoT** (e.g., **selecting tokens at each reasoning-step boundary**) into the CCoT. Specifically, we selected those relatively important tokens, which correspond to potential reasoning steps ending position, and align the synthetic CCoT with these tokens. And then we use these synthetic CCoT to fine-tune LLM to learn CCoT. The corresponding experimental results are presented in the following table:
>
> || Average Acc $\Uparrow$ | Average Len $\Downarrow$ | Average  Rel-G $\Uparrow$ |
> |-|-|-|-|
> |**Synthetic CCoT Generation via Last-Token and Answer Loss (Ours)**|**50.3**|**584.9**|**9.14**|
> |Synthetic CCoT Generation via Additional Multi-Tokens|48.6 ($\downarrow$ 1.7)|898.3 ($\uparrow$ 313.4)|5.74 ($\downarrow$ 3.40)|
>
> As shown in the results above, generating synthetic CCoT by **distilling additional multi-token information** from the DCoT **leads to reduced accuracy and more redundant generation**. The primary reason is that selecting only part of the tokens **introduces semantic inconsistency** when constructing the synthetic CCoT, which **substantially degrades its quality**. Therefore, we **choose to apply alignment only on the most informative last token, rather than incorporating inconsistent intermediate tokens**, and this design yields better downstream performance.

---

> ### Author Response · Authors · 2025-11-17
> **Authors' Response (2/4)**
>
> > W2.2: The reason why iteratively refine is needed is not clearly stated in the text.
>
> Thank you for the question. The motivation for introducing the **iteratively refinement mechanism is inspired by those prior works [1-4]**. As discussed in **Lines 209–218, [1–4] have shown that iterative refinement can improve LLM reasoning** by allowing the model to self-correct and progressively enhance its initial CCoT.
>
> To further validate its necessity in our framework, we also conducted **ablation experiments comparing our method with a variant that removes the refinement process** (in **Table 1 and Lines 372–373**). For clarity, we restate the key results here:
>
> || Average Acc $\Uparrow$ | Average Len $\Downarrow$ | Average  Rel-G $\Uparrow$ |
> |-|-|-|-|
> |**SynAdapt (Ours)**|**50.3**|**584.9**|**9.14**|
> |w/o  Refinement Process|45.6 ($\downarrow$ 4.7)|852.9 ($\uparrow$ 268.0)|5.68 ($\downarrow$ 3.46)|
>
> As shown in the results above, **our method achieves both higher accuracy and better efficiency compared with the variant without the refinement process**. This provides strong evidence that the refinement component is essential, which can effectively **strengthen LLM’s reasoning capability** and enables the **generation of higher-quality CCoT** for downstream reasoning.
>
> [1]. Yang L, Lee K, Nowak R, et al. Looped transformers are better at learning learning algorithms. arXiv. 2023.11.
> [2]. Zhu R J, Wang Z, Hua K, et al. Scaling Latent Reasoning via Looped Language Models. arXiv. 2025.10.
> [3]. Saunshi N, Dikkala N, Li Z, et al. Reasoning with latent thoughts: On the power of looped transformers. arXiv. 2025.02.
> [4]. Yu Q, He Z, Li S, et al. Enhancing auto-regressive chain-of-thought through loop-aligned reasoning. arXiv. 2025.02.
>
>
> > W2.3: The hidden dimensions of Zsyn are not marked
>
> You may have overlooked some important implementation details **provided in Appendix C.3**. As stated in Appendix C.3, **the length of the synthetic CCoT $Z_{syn}$** used in our experiments **is fixed to $m=512$**, and **its dimensionality matches the hidden state dimension of the used LLM** (e.g., **3584** for the DeepSeek-R1-Distill-Qwen-7B model). This ensures that $Z_{syn}$ is fully compatible with the model’s representation space during reasoning.
>
>
> > W3.1:  The authors did not conduct experiments on the sensitivity of the length of the synthetic CCoT, denoted as m.
>
> You may have overlooked some key experimental results in our paper. In fact, we have **conducted an in-depth hyperparameter analysis** on both the **synthetic CCoT length $m$** and refinement steps $k$, as **reported in Lines 461–464 and Appendix G**. Specifically, **Figure 7** evaluates our method under different synthetic CCoT lengths, and the results consistently **demonstrate that** **our approach remains effective across a wide range of $m$ values**.

---

> ### Author Response · Authors · 2025-11-17
> **Authors' Response (3/4)**
>
> > W3.2: Moreover, since m is set to a fixed value, I wonder whether this is appropriate for chain-of-thought reasoning, whose information content can vary greatly across different problems.
>
> Thanks for your question. In our framework, we fix the CCoT length $m$ to 4 and route hard questions to the DCoT reasoning path rather than dynamically increasing the CCoT length $m$. This is because we believe that **hard questions cannot be solved simply by extending the CCoT length**. Instead, **they require deeper reasoning via DCoT**. Moreover, **increasing the CCoT length tends to make the LLM generate more redundant content**, which have been demnonstrated in Lines 1103–1120 of our paper.
>
> Furthermore, we also **conducted additional experiments by dynamically extending the CCoT length $m$**. Specifically, we routed those mid-hard questions (0.4 < hard score < 0.7) to a longer CCoT reasoning path ($m = 6/8$) and directly predicted the answer, while remaining extremely hard questions (hard score > 0.7) to the DCoT path. The results are shown in the table below:
>
> || Acc on Easy Datasets (GSM8K, MATH500) $\Uparrow$ | Len on Easy Datasets (GSM8K, MATH500) $\Downarrow$ | Rel-G on Easy Datasets $\Uparrow$ | Acc on Hard Datasets (AMC23, AIME24/25) $\Uparrow$ | Len on Hard Datasets (AMC23, AIME24/25) $\Downarrow$ | Rel-G on Hard Datasets $\Uparrow$ |  Avg Rel-G $\Uparrow$ |
> |-|-|-|-|-|-|-|-|
> |**Fix CCoT Length $m$ at 4 （Ours）**|84.12|**903.15**|**2.63**|**58.90**|7122.67|**1.53**|**1.58**|
> |Dynamic  Extend $m$ to 6 |84.83|1077.19|2.23|37.78|**6203.61**|1.12|1.45|
> |Dynamic Extend $m$ to 8 |**84.91**|1097.07|2.18|38.61|6210.56|1.15|1.46|
>
> Here we mark the best results in **bold**.
> As illustrated in the results, for **mid-hard questions in AMC23/AIME24/AIME25**, which are originally solved via DCoT, simply increasing the CCoT length fails to solve them and even **leads to severe performance degradation**. **For mid-hard questions in GSM8K and MATH500**, which are initially solved using CCoT of length 4, although longer CCoT slightly improves accuracy, the response length also grows substantially, **resulting in a suboptimal Rel-G score** compared with our method. This is because longer CCoT increases the risk of redundant and unnecessary generation.
>
> Therefore, we conclude that **fixing a CCoT of length 4 is sufficient for easy questions**, **while hard questions should be handled via DCoT** rather than only increasing the CCoT length. **This design of our method yields the best average Rel-G score of 1.58**, as shown in the above table.
>
>
> > W4: Generalization beyond mathematical reasoning
>
> Some important experimental results may have been overlooked. **In Section 4.4 and Table 4**, we have **evaluated the generalization of our method to other domains**, including **Scientific QA (GPQA-Diamond)** and **Coding (HumanEval)**. As shown in Table 4, **our method consistently achieves a better accuracy–efficiency trade-off** for efficient reasoning compared to other baselines, which strongly demonstrates its generalization ability.

---

> ### Author Response · Authors · 2025-11-17
> **Authors' Response (4/4)**
>
> > W5: Using token count or sequence length as an efficiency metric is unfair.  ... Therefore, inference time would be a fairer basis for comparison.
>
> Thanks for your question. We have conducted additional experiments to **measure the average inference time (latency)** of our method and all baselines. We also separately **report the latency of each component in our method**, including **the refinement process** (iterative CCoT refinement + difficulty estimation) and **the autoregressive generation process** for producing the final answer. The detailed results are presented below:
>
> ||Average Generation Length per Query $\Downarrow$|Average Latency per Query (s) $\Downarrow$|
> |-|-|-|
> |Raw Model|7786.8|19.69|
> |Coconut|709.30|6.55|
> |CompressCoT|861.3|7.14|
> |CODI|941.3|7.67|
> |**SynAdapt**|**584.9**|**6.53**|
> |**=> [Refinement Process of SynAdapt]** : Iterative Refine + Difficulty Estimation|-|0.92 (**14%**)|
> |**=> [Autoregressive Generation Process of SynAdapt]**|-|5.61 (**86%**)|
>
> As shown in the results above, **our method achieves lower overall latency** compared to other baselines, demonstrating superior efficiency. This is primarily because **the time consumed by the iterative refinement and difficulty estimation** accounts for only a small proportion of the total cost (**approximately 14%**), while **the dominant factor is the autoregressive generation process**. Benefiting from the substantially reduced generation length, **our method consequently achieves the shortest overall latency**.
>
>
> ----
> We sincerely appreciate your time and effort in reviewing our paper.
> We have mark the key points of our response in **bold** for easier reading.
> **We look forward to your further feedback on our paper**. If you have any additional questions, we would be happy to discuss them!
> **If our reponses can address your concerns, can you consider to raise you score?** We will deeply appreciate for it!
> Once again, thank you very much for your thoughtful and constructive review.

---

> ### Author Response · Authors · 2025-11-23
> **Kind Remind**
>
> Dear reviewer:
>
> This is a polite reminder that we are awaiting your feedback on our rebuttal. To assist you, we have highlighted the key points for your convenience. We would greatly appreciate it if you could discuss them with us at your earliest convenience.

---

### Official Review · Reviewer_Pgav · 2025-10-30

**Soundness:** 2
**Presentation:** 2
**Contribution:** 1
**Rating:** 2
**Confidence:** 4

**Summary:**

The paper proposes an efficient reasoning framework that replaces long discrete Chain-of-Thought (DCoT) with a compact continuous CoT (CCoT) plus adaptive routing. First, it optimizes a synthetic CCoT per question while lightly aligning hidden states with the DCoT endpoint. Then it fine-tunes the LLM with LoRA to iteratively refine a draft CCoT to match the synthetic target, teaching the model to think in latent space without autoregressively emitting CoT tokens. A difficulty classifier, conditioned on the question and the refined CCoT, routes “hard” items to re-think with discrete CoT and lets “easy” ones answer directly from CCoT. Across math, scientific QA, and coding, SynAdapt reports shorter generations with competitive accuracy.

**Strengths:**

The authors implement wide experiments to evaluate the performance of their method.The proposed method is empirically shown to be effective in reasoning tasks in the author’s setting. The authors compare their method with various baselines on different datasets.

**Weaknesses:**

This paper introduces a quite complex framework based on different components proposed by past works, while it lacks really interesting or important insights/ findings.

The effectiveness of the framework may be questionable. Based on Table 1, when fully using CCoT, the performance is basically the same as directly prompting the model to give the output. The better performance in the accuracy-sensitive scenario is because the full CoT of the model is used. So this naturally questions why we ever need the proposed framework? A difficulty estimator is good enough if it can decide when to directly output the answer and when to use full CoT.

The experiments are only done on a single model. It remains unclear whether the framework is ad-hoc to that model.

There are many parameters to tune to make the framework work, for example, the number of iterations in refinement.  The refinement process also takes time, which is not taken into consideration for efficiency.

The paper implies good CCoT should be distilled step by step from DCoT (this assumption is not very intuitive and lacks evidence). However, the framework itself is not distilling each step from DCoT but last token (which is similar to CoDI). This is a bit against the advantages they claimed in their framework.  The motivation for Synthetic CCoT is not strong or maybe this is because of writing.

**Questions:**

How do you decide which baselines are for accuracy-sensitive
scenario or efficiency-sensitive scenario? The current division does not make sense to me. For example, tokenbudget is designed to be efficient.  Isn’t accuracy-sensitive scenario for reasoning models that take super long reasoning time to get superior performance?

What is the length here? Is it the number of tokens?

What is difference between CoT-SFT and raw model? How do you do SFT here?

What are some particular reasons for refinement? Is it better than directly optimizing  toward the target CCoT?

---

> ### Author Response · Authors · 2025-11-17
> **Authors' Response (1/5)**
>
> We sincerely thank you for your feedback and suggestions! There exist many misunderstandings. We address all your concerns below:
>
> > W1: This paper introduces a quite complex framework based on different components proposed by past works, while it lacks really interesting or important insights/ findings.
>
> Indeed, there are some misunderstandings about the actual contribution of our method. Our method consists of two critical component: **thinking via CCoT** and **judging difficluty via CCoT**. While these components build upon previous work, they both introduce novel designs tailored to address the specific challenge.
>
> For the **thinking via CCoT** component, traditional CCoT-based methods such as Coconut, CODI, and CompressCoT suffer from suboptimal fine-tuning performance due to **imperfect alignment targets**, as illustrated in Figure 1. To overcome this limitation, we introduce **synthetic CCoTs, which are generated before fine-tuning and used as enhanced alignment targets**. These synthetic CCoTs effectively capture reasoning information, thereby achieving better fine-tuning performance. As demonstrated in Table 1, this design allows our method to substantially outperform all other CCoT-based approaches.
>
> For the **judging difficulty via CCoT** component, traditional methods (e.g., Seq_PPL, PromptLLM, RouteLLM, and Probe_Q) **only rely on superficial cues** from the question to judge difficulty, such as perplexity and lexical patterns (Lines 399–406). However, as shown in Figure 6, difficult questions may appear similar to easy ones, sharing short length and similar lexical structures. Relying solely on such superficial cues is insufficient. To overcome this limitation, we **introduce a difficulty classifier that leverages both the question and the reasoning information encoded in the CCoT**. As demonstrated in Table 2, incorporating additional information from CCoTs enables our method to **more effectively identify hard questions**. Furthermore, due to this improved identification capability, our method achieves **better downstream task performance** compared to other hard-question identification baselines, as shown in Figures 3(c) and 3(d).

---

> ### Author Response · Authors · 2025-11-17
> **Authors' Response (2/5)**
>
> > W2.1:  Based on Table 1, when using CCoT, the performance is basically the same as directly prompting the model to give the output.
> W2.2: The better performance in the accuracy-sensitive scenario is because the CoT of the model is used.
>
> There are some misunderstandings about our experiment results. Firstly, **Using CCoT corresponds to the** "***SynAdapt (τ = 1.0)*** " row in Table 1, while **Directly Prompting to Output corresponds to the "*NoThinking*"** row. **The performance of these two methods differs significantly**, where Using CCoT is for efficiency-sensitive settings, whereas Directly Prompting to Output is for accuracy-sensitive ones. Here, we repeat some results from Table 1 for clarity:
>
> || Acc on Easy Questions (GSM8K, MATH500) $\Uparrow$ | Len on Easy Questions (GSM8K, MATH500) $\Downarrow$ | Rel-G on Easy Questions $\Uparrow$ | Acc on Hard Questions (AMC23, AIME24/25) $\Uparrow$ | Len on Hard Questions (AMC23, AIME24/25) $\Downarrow$ | Rel-G on Hard Questions $\Uparrow$ |  Avg Rel-G $\Uparrow$ |
> |-|-|-|-|-|-|-|-|
> |**Accuracy-Sensitive Scenario**||||||||
> |Using CoT|91.95|2599.10|1.00|60.83|11244.67|1.00|1.00|
> |Mix CCoT with CoT (**Ours**)|84.12|903.15|2.63|58.90|7122.67|**1.53**|**1.58**|
> |Directly Prompt to Output|84.05|792.45|**2.99**|48.33|9074.10|0.98|1.21|
> |**Efficiency-Sensitive Scenario**||||||||
> |Using CCoT (**Ours**)|82.05|496.45|**4.68**|29.16|643.80|**8.37**|**9.14**|
>
> We mark the best Rel-G score for both scenarios in **bold**.
> And as shown in the above table, **for easy questions**, Using CCoT **achieves more effective length reduction** and a better accuracy–efficiency trade-off, as indicated by its **higher Rel-G score (4.68)** compared to Directly Prompt to Output (2.99). **For hard questions**, Directly Prompt to Output **fails to substantially reduce reasoning length**, whereas **Using CCoT performs consistently better**. This limitation arises because prompt engineering alone cannot effectively constrain the reasoning process, and LLMs tend to generate redundant reasoning process for complex problems, which is also demonstrated by [1,2].
>
> Secondly, **the strong performance of our method** (Mix CCoT with CoT, i.e., *"SynAdapt (τ = 0.5)"* row in Table 1) **in accuracy-sensitive scenarios should be attributed to the mixed usage of CCoT and CoT**, **rather than relying solely on CoT**. As shown in above table, compared to the Using CoT baseline, our method (Mix CCoT with CoT) can **offer better accuracy-efficiency trade-off, achieving better Rel-G score of 1.58**. **For easy questions**, as shown in Figure 3(b), our method identifies most of them and directly uses CCoT to generate answers, **skipping the generation of unnecessarily long CoTs**. **For hard questions**, our method concatenates the question with its corresponding CCoT before generating the subsequent CoT, thereby **skipping redundant reasoning steps** and reducing the overall reasoning length. Therefore, our method can realize better trade-off performance compared to Using CoT.
>
> [1]. Sun Y, Wang H, Li J, et al. An Empirical Study of LLM Reasoning Ability Under Strict Output Length Constraint. arXiv. 2025.04.
> [2]. Nayab S, Rossolini G, Simoni M, et al. Concise thoughts: Impact of output length on llm reasoning and cost. arXiv. 2024.07.

---

> ### Author Response · Authors · 2025-11-17
> **Authors' Response (3/5)**
>
> > W2.3: A difficulty estimator is good enough if it can decide when to directly output the answer and when to use CoT. Why do we ever need the proposed framework?
>
> You may have overlooked some important details about our framework. **First of all**, our framework is necessary because **traditional difficulty estimators** (e.g., Seq_PPL, PromptLLM, and so on) **rely solely on superficial cues** (e.g., perplexity or lexical patterns) and fail to capture the underlying reasoning complexity of questions. **Our framework addresses this limitation by incorporating CCoT information during difficulty estimation**, enabling the better difficulty estimation performance. As shown in Table 2, **this design leads to significantly more accurate difficulty estimation**, compared other estimator baselines. Furthermore, due to the improved estimation quality, **our framework achieves superior downstream task performance** compared to all baselines, as illustrated in Figures 3(c) and 3(d). Therefore, the proposed framework is essential for effectively integrating additional CCoT information into difficulty estimation and achieving better downstream performance.
>
> **Secondly**, our framework performs well because the **Using CCoT for Easy Questions provides a better accuracy–efficiency trade-off compared to Directly Prompt to Output** for easy questions, as demonstrated in our responses to W2.1 and W2.2.
>
> In addition, we conducted further experiments to compare the baseline that use the difficulty estimator without CCoT (i.e., **Estimator without CCoT + Direct Output for Easy Questions + CoT for Hard Questions**) with our proposed framework (i.e., **Estimator with CCoT + CCoT for Easy Questions + CoT for Hard Questions**) in the following table:
>
> || Acc on Easy Datasets (GSM8K, MATH500) $\Uparrow$ | Len on Easy Datasets (GSM8K, MATH500) $\Downarrow$ | Rel-G on Easy Datasets $\Uparrow$ | Acc on Hard Datasets (AMC23, AIME24/25) $\Uparrow$ | Len on Hard Datasets (AMC23, AIME24/25) $\Downarrow$ | Rel-G on Hard Datasets $\Uparrow$ |  Avg Rel-G $\Uparrow$ |
> |-|-|-|-|-|-|-|-|
> |Estimator without CCoT + Direct Output for Easy + CoT for Hard|**87.68**|1452.22|1.71|51.11|10394.10|0.91|1.02|
> |**Estimator with CCoT + CCoT for Easy + CoT for Hard (Ours)**|84.12|**903.15**|**2.63**|**58.90**|**7122.67**|**1.53**|**1.58**|
>
> Here we mark better results in **bold**.
> As shown by the results, **for the easy datasets** (GSM8K and MATH500), our method incurs only a slight accuracy drop while **achieving a substantial reduction in generation length**, leading to the best Rel-G score of 2.63. **For the hard datasets** (AMC23 and AIME24/25), our method **simultaneously achieves higher accuracy and shorter generation length**. This improvement primarily stems from the **more accurate identification of hard questions** by incorporating CCoT into the difficulty estimation.
> **Overall, by introducing CCoT, our framework achieves a superior average Rel-G score of 1.58, demonstrating a better accuracy–efficiency trade-off compared to only using an estimator without CCoT.**
>
> > W3: The experiments are only done on a single model.
>
> Some important parts of our paper may have been overlooked. We conducted extensive experiments to **evaluate the generalization ability of our method across different LLMs (Llama-8B and Qwen-1.5B) in Section 4.4 and Table 5**, which also addresses our research question 4 stated in Lines 280–282. As shown by these experiment results, **our SynAdapt consistently outperforms all baseline methods across these LLM backbones**, strongly demonstrating the robustness and generalization capability of our approach.
>
> > W4.1: There are many parameters to tune to make the framework work, for example, the number of iterations in refinement
>
> Indeed, you may overlook some important experiments in our paper. We **have conducted extensive hyperparameter analyses on** **the number of refinement iterations ($k$)** and the CCoT length ($m$), **as described in Lines 462–464 of Section 4.4, Figures 7 and 8, and Appendix G**. We also provide a detailed discussion of how these hyperparameters affect our framework in Appendix G.
>
> Here we briefly repeat the influence of the number of refinement iterations $k$ again. **As $k$ increases, the CCoT becomes progressively refined, leading to improved accuracy**. However, **when $k$ exceeds 4, redundant refinement steps may confuse the LLM**, resulting in longer generation lengths and a slight drop in accuracy. For more detailed analyses of these parameters, please refer to Appendix G.

---

> ### Author Response · Authors · 2025-11-17
> **Authors' Response (4/5)**
>
> > W4.2: The refinement process also takes time, which is not taken into consideration for efficiency.
>
> Thank you for your question. We agree that the refinement process also incurs computational cost, and thus **the total generation time (latency)** should be evaluated in addition to generation length. To address this, we conducted additional experiments that measured **both latency and generation length** for our method and the baselines. The results are summarized in the following table:
>
> ||Average Generation Length per Query $\Downarrow$|Average Latency per Query (s) $\Downarrow$|
> |-|-|-|
> |Raw Model|7786.8|19.69|
> |Coconut|709.30|6.55|
> |CompressCoT|861.3|7.14|
> |CODI|941.3|7.67|
> |**SynAdapt**|**584.9**|**6.53**|
> |**=> [Refinement Process of SynAdapt]**|-|0.92 (**14%**)|
> |**=> [Autoregressive Generation Process of SynAdapt]**|-|5.61 (**86%**)|
>
> Here we mark better results in **bold**.
> As shown in the results above, **the time required for our refinement process is relatively small**. This is because we adopt an iterative refinement strategy with **only four refinement steps**, rather than generating the entire CCoT in an autoregressive manner (see Figure 2 and Lines 214–233). This design is highly efficient and **the total number of LLM forward passes equals the number of refinement iterations (i.e., four)**. As a result, the refinement process accounts for **only 14% of the total latency** of our method. Moreover, our method achieves the **lowest overall latency**, primarily because it substantially shortens the generation length during the reasoning process. This further demonstrates the efficiency advantages of our approach.
>
> > W5: The paper implies good CCoT should be distilled step by step from DCoT. However, the framework itself is not ... but the last token. This is a bit against the advantages they claimed in their framework.
>
> Firstly, there may be some misunderstanding about our implication and contribution. We do not claim that a good CCoT should be distilled step by step from DCoT. **Our actual claim is that, to fine-tune an LLM to learn CCoT, it is crucial to establish a better alignment (training) target**. As described in Figure 1 and Lines 51–71 of the Introduction, existing CCoT-based methods (e.g., Coconut, CODI, and CompressCoT) suffer from suboptimal alignment objectives when training LLMs to learn CCoT. This leads to indirect, single, or inconsistent alignment targets. Therefore, **our main advantage lies in generating synthetic CCoT before fine-tuning**, which serve as improved full-alignment targets and enable more effective learning of CCoT, as illustrated in Figure 1.
>
> Secondly, when generating the synthetic CCoT as the alignment target, our method **not only distills information from the last token** of DCoT to form the synthetic CCoT, **but also introduces an additional answer loss**, as shown in Equation (1). This loss is designed to optimize the synthetic CCoT **such that it encodes sufficient reasoning information to derive the correct answer**.
>
> Thirdly, we also conduct experiment to **compare our method with distilling multiple tokens (i.e., at each reasoning step) from DCoT to CCoT**. Specifically, we selected relatively important tokens, those corresponding to potential reasoning steps, from DCoT and distilled them into the synthetic CCoT, which was then used to fine-tune the LLM to learn CCoT. The corresponding experimental results are presented in the following table:
>
> || Average Acc $\Uparrow$ | Average Len $\Downarrow$ | Average  Rel-G $\Uparrow$ |
> |-|-|-|-|
> |**Synthetic CCoT Generation via Last-Token and Answer Loss (Ours)**|**50.3**|**584.9**|**9.14**|
> |Synthetic CCoT Generation via Additional Multi-Tokens|48.6 ($\downarrow$ 1.7)|898.3 ($\uparrow$ 313.4)|5.74 ($\downarrow$ 3.40)|
>
> Here we mark better results in **bold**.
> As shown in the above results, generating synthetic CCoT by **distilling additional multi-token** from the DCoT **leads to decreased accuracy and more redundant generation**. The primary reason is that selecting only a small subset of tokens from the DCoT introduces **semantic inconsistency**, in other words, **a few isolated tokens cannot fully capture the complete reasoning process encoded in the original DCoT**. This is similar to the limitations observed in CompressCoT, as illustrated in Figure 1 and Lines 67–71. Consequently, **the synthetic CCoT becomes lower in quality**, leading to degraded downstream performance and a **suboptimal accuracy–efficiency trade-off** compared with our method.

---

> ### Author Response · Authors · 2025-11-17
> **Authors' Response (5/5)**
>
> > Q1: How do you decide which baselines are for ...? Isn’t accuracy-sensitive scenario ... take super long reasoning time to get superior performance?
>
> Indeed, there may be some misunderstanding about the definitions of our accuracy-sensitive and efficiency-sensitive scenarios. First of all, we should emphasize that **both scenarios share the same key overarching premise: enabling efficient reasoning for LLMs**. As stated in Lines 866–869 and Lines 893–895, **the difference between both scenarios lies in their tolerance for accuracy loss**. The accuracy-sensitive scenario requires maintaining accuracy comparable to the raw model while reducing the generation length. In contrast, the efficiency-sensitive scenario allows a moderate accuracy drop in exchange for substantial reductions in reasoning length. However, regardless of the scenario, the core requirement remains the same: **the selected baselines must reduce overall reasoning length and latency**. Therefore, approaches that take longer reasoning time than the raw model, even if they achieve higher accuracy, **violate this fundamental premise of efficient reasoning**, and thus are not included as baselines in our comparison.
>
> Secondly, we also **provide detailed explanations on how we categorize each baseline** into the accuracy-sensitive or efficiency-sensitive scenario **in Lines 303–322 and Appendix C.2**. Briefly, the key criterion is **whether the baseline introduces a substantial accuracy drop**. Methods that incur notable accuracy degradation are categorized under the efficiency-sensitive scenario, whereas those that maintain accuracy comparable to the raw model fall into the accuracy-sensitive scenario.
>
>
> > Q2: What is the length here?
>
> As stated in Line 293 of our paper, the “length” refers to **the total number of tokens generated during both the reasoning process and the final answer**. Since LLMs generate tokens in an autoregressive manner, **the total number of generated tokens is directly correlated with the latency** required to respond to each user query.
>
> > Q3: What is difference between CoT-SFT and raw model? How do you do SFT here?
>
> As described in Lines 306–307 and Lines 870–871, CoT-SFT fine-tunes the LLM using **both the CoT and corresponding answer from the training set** through supervised fine-tuning (SFT). The CoTs are generated by expert LLMs (e.g., DeepSeek-R1) and are often **concise while retaining high-quality reasoning**. Fine-tuning on this data helps the model **maintain accuracy while producing shorter, more efficient reasoning steps**. In contrast, the raw model directly produces answers without fine-tuning, often resulting in longer and less efficient reasoning.
>
> > Q4: What are some particular reasons for refinement? Is it better than directly optimizing toward the target CCoT?
>
> Indeed, the introduction of the refinement mechanism for generating CCoT **is inspired by recent works [1–4]**, as stated in Lines 209–218. **These studies demonstrate that iterative refinement can substantially enhance LLM reasoning** by enabling the model to repeatedly verify, adjust, and improve its initial CCoT. Furthermore, we also conducted ablation studies to directly **compare our method with its variant without the refinement process (Table 1 and Lines 372–373)**. For clarity, we restate the key results here:
>
> || Average Acc $\Uparrow$ | Average Len $\Downarrow$ | Average  Rel-G $\Uparrow$ |
> |-|-|-|-|
> |**SynAdapt (Ours)**|**50.3**|**584.9**|**9.14**|
> |w/o  Refinement Process|45.6 ($\downarrow$ 4.7)|852.9 ($\uparrow$ 268.0)|5.68 ($\downarrow$ 3.46)|
>
> Here we mark better results in **bold**.
> As shown in the results above, **our method achieves both higher accuracy and better efficiency compared with the variant without the refinement process**. This provides strong evidence that the refinement component is essential, which can effectively **strengthen LLM’s reasoning capability** and enables **the generation of higher-quality CCoT** for downstream reasoning.
>
> [1]. Yang L, Lee K, Nowak R, et al. Looped transformers are better at learning learning algorithms. arXiv. 2023.11.
> [2]. Zhu R J, Wang Z, Hua K, et al. Scaling Latent Reasoning via Looped Language Models. arXiv. 2025.10.
> [3]. Saunshi N, Dikkala N, Li Z, et al. Reasoning with latent thoughts: On the power of looped transformers. arXiv. 2025.02.
> [4]. Yu Q, He Z, Li S, et al. Enhancing auto-regressive chain-of-thought through loop-aligned reasoning. arXiv. 2025.02.
>
>
> ----
> We sincerely appreciate your time and effort in reviewing our paper.
> We have mark the key points of our response in **bold** for easier reading.
> **We look forward to your further feedback on our paper**. If you have any additional questions, we would be happy to discuss them!
> **If our reponses can address your concerns, can you consider to raise you score?** We will deeply appreciate for it!
> Once again, thank you very much for your thoughtful and constructive review.

---

> ### Author Response · Authors · 2025-11-23
> **Kind Remind**
>
> Dear reviewer:
>
> This is a polite reminder that we are awaiting your feedback on our rebuttal. To assist you, we have **highlighted the key points** for your convenience. We would greatly appreciate it if you could discuss them with us at your earliest convenience.

---

> ### Comment · Reviewer_Pgav · 2025-11-25
>
> I truly appreciate the authors' response and efforts. But I tend to maintain my score as my questions have not been addressed.
>
> Some major questions are:
>
> 1. The paper repeats "alignment'' target for CCoT and considers that past work "suffer from suboptimal alignment objectives when training LLMs to learn CCoT. This leads to indirect, single, or inconsistent alignment targets".
>
>  I think "alignment" is a rather abstract and uncommon term here. The authors have clarified that good alignment is not about distilling step-by-step, although the figure 1 (single alignment vs full alignment) and the writing may indeed give readers that impression. What defines good alignment, and why past work is bad in alignment are really unclear to me and lack concrete and solid analysis. If this is a conclusion from past work, it should be cited.
>
> 2. accuracy-sensitive vs efficiency-sensitive
>
> I still feel this is a counterintuitive setting that divides all baselines into two categories. Overall, an ideal efficient CoT method is accurate while reducing the token cost. The baselines in Table 1, even for accuracy-sensitive cases are more or less proposed for efficiency. So I find the reasons and the motivation to classify different efficient methods into two scenarios are kind of weak. Readers may not agree with some of your choices for classification.
>
> "...where Using CCoT is for accuracy-sensitive settings, whereas Directly Prompting to Output is for efficiency-sensitive ones." But in the table, you put CCoT in efficiency-sensitive and Directly Prompting to Output in the efficiency-sensitive ones. This makes me more confused.
>
> I asked "What Len means in Table 1" because I find some results counterintuitive. For example, tokenskip has very large token lengths, even larger than the raw model. This is very different from the results in the paper of Tokenskip.  Similarly, CoT-FT is proposed to reduce token length, while it turns out to produce longer CoT. The authors should double-check their results.
>
> 3. There are many parameters to tune to make the framework work.
>
> I know the authors conduct the ablation study to investigate what parameters to use, while it does not affect the fact that we need to tune parameters when using the framework.

---

> > ### Author Response · Authors · 2025-11-26
> > **Further Authors' Response (1/2)**
> >
> > We sincerely appreciate your reply and feedback! Below, we address your remaining concerns:
> >
> > > Q1:  What defines good alignment, and why past work is bad in alignment ... The authors have clarified that good alignment is not about distilling step-by-step, although the figure 1 (single alignment vs full alignment) and the writing may indeed give readers that impression. ...
> >
> > Firstly, we would like to clarify that **the term *alignment* is not introduced by our work** and **it has been widely used in prior studies**, **including the CODI baseline [1] and others [2–4]**.
> > In the context of CCoT-based reasoning, ***alignment* process refers to fine-tuning the LLM to learn how to reason via CCoT**.
> > And ***good alignment* depends critically on whether we can provide an appropriate ground-truth supervision signal**, called the ***alignment target*** in our paper, for effective CCoT learning.
> > Due to the inherent heterogeneity between CCoT and the original DCoT (e.g., differences in length and structure), directly using the raw DCoT as the alignment target is infeasible.
> > Therefore, we argue that **identifying a high-quality alignment target is crucial** for improving CCoT-based reasoning performance, **which is precisely the key problem addressed in our paper**.
> >
> > Secondly, **the reason of why past works is bad in alignment is mainly due to suboptimal alignment targets**, which we have clearly illustrated in **Figure 1 and Lines 50–71**.
> > Here we briefly restate again: **Coconut** does not use any explicit alignment target for CCoT, which we refer to as **indirect training**.
> > **CODI** relies only on the last token of the DCoT as the alignment target, which we refer to as **single alignment**.
> > **CompressCoT** selects a subset of tokens from the DCoT, but these partial tokens cannot maintain semantic consistency, leading to an **inconsistent alignment target**.
> > We believe that **Figure 1 and Lines 50–71 of our paper is clear enough to describe the limitations** of existing baseline methods.
> > Moreover, **our paper does not contain any statements related to this description of "*distilling step by step*"**, which only appears in your review and is likely due to a misunderstanding or hallucination.
> >
> > **Furthermore, the limitations of these baselines methods are not only identified by us, but are also acknowledged in their original works**.
> > For instance, in the Limitation section of CODI [1], the authors explicitly state:
> > ```python
> > Another limitation of the current CODI is the absence of intermediate gradients until the end of the sequence ...
> > This issue could become more pronounced when scaling to more complex problems requiring longer continuous reasoning chains.
> > ```
> > This statement also underscores that **only using the last token (the end of the sequence) is insufficient** and causes sub-optimal performance.
> > Therefore, **finding a full-sequence, semantically consistent alignment target is essential for advancing CCoT-based reasoning, which is also the core focus and contribution of our work.**
> >
> > [1]. CODI: Compressing Chain-of-Thought into Continuous Space via Self-Distillation. EMNLP 2025.
> > [2]. LLMs Know More Than They Show: On the Intrinsic Representation of LLM Hallucinations. ICLR 2025.
> > [3]. Chasing Consistency: Quantifying and Optimizing Human-Model Alignment in Chain-of-Thought Reasoning. Arxiv 2025.
> > [2]. DeAL: Decoding-time Alignment for Large Language Models. ACL 2025.
> >
> > > Q2.1 : Overall, an ideal efficient CoT method is accurate while reducing the token cost. ... I find the reasons and the motivation to classify different efficient methods into two scenarios are kind of weak.
> >
> > Indeed, the reason **why we categorize these baseline methods** into accuracy-sensitive and efficiency-sensitive scenarios **is to ensure a fair comparison between different methods**.
> > In practice, **accuracy and efficiency are unavoidable in trade-off: some methods prioritize preserving accuracy, while others emphasize extreme efficiency**.
> > Therefore, **comparing baselines from different scenarios is unreasonable and unfair**.
> > For example, CoT-FT belongs to the accuracy-sensitive scenario, whereas NoCoT-FT belongs to the efficiency-sensitive scenario.
> > **CoT-FT achieves higher accuracy** but lower efficiency, while **NoCoT-FT exhibits higher efficiency** but lower accuracy.
> > **So, it is difficult to directly compare these two methods from different scenarios and claim which one is "*better*".**
> >
> > Moreover, we also introduces a new metric, **Rel-G**, which measures the **relative efficiency gain per unit accuracy cost**.
> > This metric **provides a unified comparison standard for all baselines**, regardless of their scenario.
> > So **our paper also support unified comparison between all baselines and our method via the Rel-G score**.
> > As shown in Table 1, **our method achieves the global best Rel-G score** of 9.14 among all accuracy-sensitive and efficiency-sensitive baselines, demonstrating its overall effectiveness.

---

> > ### Author Response · Authors · 2025-11-26
> > **Further Authors' Response (2/2)**
> >
> > > Q2.2: Using CCoT is for accuracy-sensitive settings, whereas Directly Prompting to Output is for efficiency-sensitive ones.
> >
> > Thanks for your careful suggestion! This is a typo and we have revised it. Using CCoT is efficiency-sensitive and Directly Prompting to Output is accuracy-sensitive.
> >
> >
> > > Q2.3:  Tokenskip has very large token lengths, even larger than the raw model. This is very different from the results in the paper of Tokenskip.
> >
> > Although we briefly explained the phenomena in Lines 347–355, we now provide more detailed evidence to explain it.
> > **From the LLM backbone perspective**, **we focus on those reasoning LLMs** (e.g., DeepSeek-Qwen-Distill-7B), **which differs substantially with the LLM backbone** (LLaMA-3.1-8B-Instruct) **used in original TokenSkip**.
> > The reasoning LLM is obtained through multiple rounds of continual RL-tuning, and as shown in [1, 2], **more continual fine-tuning significantly increases instability of these reasoning LLMs**.
> > **The instability making those reasoning LLMs more likely to produce endless or overly lengthy outputs when TokenSkip is applied, compared to the stable instruction-tuned backbone.**
> > Therefore, we observe that applying Tokenskip to reasoning LLM will increase the generation length, even longer than the raw model.
> >
> > **From the training dataset perspective**, **the training datasets used in original TokenSkip** is from GSM8K and MATH500, **which contain relatively simple questions** with an average DCoT length of only 357 tokens.
> > Moreover, the training datasets used in original TokenSkip are **fully in-domain with their evaluation benchmarks** (also from GSM8K and MATH500).
> > Then original Tokenskip discards part of the reasoning tokens still allows the model to infer the correct answer directly and reduce the generation length.
> > **In contrast**, **the training data of our *SynAdapt* (DeepMath-103K) is significantly more difficult**, with an average reasoning length of 10,649 tokens.
> > For such complex problems, **removing parts of the reasoning chain easily disrupts crucial steps**, resulting in incorrect results, **endless and longer generation**, compared to raw model.
> > The phenomenon that imperfect fine-tuning of reasoning LLMs can lead to excessively long or even endless outputs has also been observed in concurrent work [1].
> >
> > **Therefore, we argue that TokenSkip is more suitable for only simple-question scenarios.**
> > When dealing with complex tasks or using reasoning LLMs, our method offers substantially more stable and reliable performance.
> >
> > [1]. Let's Let's Let's Let's... Understand Looping in Reasoning Models. OpenReview 2025.
> > [2]. An Empirical Study of Catastrophic Forgetting in Large Language Models During Continual Fine-tuning. ICLR 2024.
> >
> >
> > > Q2.4: Similarly, CoT-FT is proposed to reduce token length, while it turns out to produce longer CoT.
> >
> > We also provide more detailed evidence to explain the behavior of CoT-FT.
> > As discussed in our previous response and in Lines 306–307 and 870–871 of our paper, **CoT-FT continually fine-tunes the LLM backbone using those CoTs generated by expert LLMs** (e.g., DeepSeek-R1-671B).
> > The goal is to leverage **these high-quality reasoning CoTs** to help the model avoid incorrect reasoning paths and reduce unnecessary generation.
> > However, although expert LLMs can provide high-quality CoT that leads to correct answers, **their generated CoTs are typically much longer**, as shown in the table below:
> >
> > ||Len on Easy Questions (MATH500)|Len on Hard Questions ( AIME24)|
> > |-|-|-|
> > |Initial Backbone (Qwen-7B-Distill)|1110|14071|
> > |Expert LLM (Deepseek-R1-671B)|**1686** ($\uparrow$ 576) |**16970** ($\uparrow$ 2899)|
> >
> > This is because that **correctly solving certain problems requires longer and more detailed reasoning than just producing an incorrect answer**.
> > Therefore, although **CoT-FT can improve accuracy on hard questions (e.g., AIME25), it will inadvertently increase the model’s generation length**.
> > **In addition, on easier questions**, **CoT-FT still tends to produce redundant and even incorrect reasoning steps, which results in accuracy degradation** on MATH500 and GSM8K.
> >
> > > Q4: There are many parameters to tune to make the framework work.
> >
> > We emphasize that **the only parameter requiring tuning in our framework** during real-world deployment **is just the difficulty threshold $\tau$**.
> > Its value can be dynamically adjusted according to the deployment scenario: **using a lower $\tau$ in accuracy-sensitive settings and a higher $\tau$ in efficiency-sensitive ones**.
> > This makes our approach highly efficient and cost-effective for practical deployment.
> >
> >
> > ----
> >
> > We sincerely appreciate your efforts.
> >
> > **We look forward to your further feedback on our paper. If you have any additional questions, we would be happy to discuss them!**

---

### Official Review · Reviewer_Kakc · 2025-10-31

**Soundness:** 3
**Presentation:** 3
**Contribution:** 3
**Rating:** 6
**Confidence:** 3

**Summary:**

The paper introduces SynAdapt, a framework that synthesises CCoT for a target reasoning LLM, where a adapter LLM is trained iteratively to predict a good CCoT needed for reasoning problem solving. The authors show high sequence length efficiency while preserving most accuracy.

**Strengths:**

1. Novel, tightly-coupled design: The three-stage pipeline (find optimal CCoT for LLM A → train LLM B to mimic it → deploy B+A) is novel and interesting.

2. The authors conduct extensive experiments to prove the superiority of SynAdapt against baselines.

**Weaknesses:**

> Compute cost is only partially accounted for

Training LLM B requires n iterations and each forward pass concatenates the full current CCoT (length m) with the question.  Complexity O(n*m) is paid during synthesis, yet Table 1 quotes only the final inference length.  If n≈4 and m≈512, the total FLOPs *before seeing a single test example can be already large.  A FLOPs count that includes the iterative stage is needed to argue for true efficiency.

> Performance concerns

Although high token efficiency, the accuracy drop in Table 1 is equally large. Since the “length” column only counts the final tokens for reasoning parts, any claimed trade-off between accuracy and sequence length is meaningless.

**Questions:**

See Weaknesses.

---

> ### Author Response · Authors · 2025-11-17
> **Authors' Response (1/1)**
>
> We sincerely thank you for your feedback and suggestions! We address all your concerns below:
>
> > W1: Compute cost is only partially accounted for. Complexity O(n*m) is paid during synthesis.
> W3: Since the “length” column only counts the final tokens for reasoning parts, ... is meaningless.
>
> Thanks for your question. We believe there may still be some misunderstanding regarding the efficiency of our method. Firstly, **the computational complexity of our approach is O(n) rather than O(n·m)**. As shown in Figure 2 and Lines 214–233, our method synthesizes the CCoT by **starting from a randomly initialized draft and iteratively refining it**, rather than autoregressively generating every token. This design is highly efficient, which **only requires n forward passes of the LLM**, where n is the number of refinement iterations. In our implementation, n=4, so **the total computational cost is merely O(4 × one-forward FLOPs), which is extremely small**.
> Secondly, the "length" metric refers to **the total number of tokens in both the reasoning and answer segments**. These tokens **must be autoregressively generated** by the LLM, and **autoregressive decoding is the dominant contributor to total latency** during LLM generation. Thus, **the length of generated tokens directly reflects the LLM’s runtime** and **can serves as a reliable indicator of the efficiency** of different methods.
> Furthermore, we also measured the total generation time (latency) of our method and the proportion of time spent in each process in the following table:
>
>
> ||Average Generation Length per Query $\Downarrow$|Average Latency per Query (s) $\Downarrow$|
> |-|-|-|
> |Raw Model|7786.8|19.69|
> |Coconut|709.30|6.55|
> |CompressCoT|861.3|7.14|
> |CODI|941.3|7.67|
> |**SynAdapt**|**584.9**|**6.53**|
> |**=> [Refinement Process of SynAdapt]**|-|0.92 (**14%**)|
> |**=> [Autoregressive Generation Process of SynAdapt]**|-|5.61 (**86%**)|
>
> We marked better results in **bold**.
> And as shown in the results above, **our method achieves the lowest average generation latency** among all baselines, clearly demonstrating its efficiency. Moreover, we observe that the **autoregressive generation process accounts for the majority of the total latency (86%)**, which further supports that the **generation length is a reliable proxy for measuring the efficiency** of different methods.
>
> > W2: Although high token efficiency, the accuracy drop in Table 1 is equally large.
>
> Thanks for your question. There may be some misunderstanding regarding the accuracy performance of our method in Table 1. For Table 1, we have considered an important efficient reasoning scenario: the **accuracy-sensitive scenario**, which aims to maintain accuracy comparable to the raw model while reducing the generation length. Our method performs effectively in this scenario. As shown in Table 1, in the **accuracy-sensitive scenario, our method maintains nearly comparable accuracy while significantly reducing the generation length**. For clarity, we restate the key results below:
>
> || Average Acc $\Uparrow$ | Average Len $\Downarrow$ | Average  Rel-G $\Uparrow$ |
> |-|-|-|-|
> |Raw Model|**73.3**|7786.8|1.00|
> |**SynAdapt when $\tau=0.5$ (Ours)**|69.0 ($\downarrow$ 4.3)|**4694.8** ($\downarrow$ 3092.04)|**1.58** ($\uparrow$ 0.58)|
>
> We marked better results in **bold**.
> And as shown above table, our method **reduces the average generation length by nearly 50%** (from 7786.84 to 4694.80) while **maintaining comparable accuracy, with only a minor drop of 4.3 points**. This demonstrates that our method can **achieve high token efficiency with minimal accuracy drop**, which is highly valuable for practical applications.
>
> ----
> We sincerely appreciate your time and effort in reviewing our paper.
> We have mark the key points of our response in **bold** for easier reading.
> **We look forward to your further feedback on our paper**. If you have any additional questions, we would be happy to discuss them!
> **If our reponses can address your concerns, can you consider to raise you score?** We will deeply appreciate for it!
> Once again, thank you very much for your thoughtful and constructive review.

---

### Note · Authors · 2025-12-16

I have read and agree with the venue's withdrawal policy on behalf of myself and my co-authors.